# 1 Impacts of shipping emissions on ozone pollution in China

- 2 Zhenyu Luo<sup>#,1</sup>, Li Peng<sup>#,1</sup>, Zhaofeng Lv<sup>1</sup>, Junchao Zhao<sup>2</sup>, Tingkun He<sup>1</sup>, Wen Yi<sup>1</sup>, Yongyue
- 3 Wang<sup>1</sup>, Kebin He1, Huan Liu\*,1
- <sup>1</sup>State Key Laboratory of Regional Environment and Sustainability, School of Environment,
- Tsinghua University, Beijing 100084, China
- <sup>2</sup>Key Laboratory of Vehicle Emission Control and Simulation of Ministry of Ecology and
- Environment, Vehicle Emission Control Center, Chinese Research Academy of environmental
- Sciences, Beijing, China
- # These authors contributed equally to this work.
- \* Correspondence: Huan Liu (liu env@tsinghua.edu.cn)

1112

13

14

15

1617

18

1920

2627

#### Abstract

With the Two Phases of Clean Air Actions in China, the shipping sector has emerged as a significant source with substantial emission reduction potential compared to land-based anthropogenic sectors. Therefore, understanding the contribution of shipping emissions to ozone (O<sub>3</sub>) pollution is therefore essential for advancing China's air pollution control efforts. In this study, a coupled framework including a chemical transport model with machine learning techniques was developed to systematically investigate the interannual and seasonal impacts of shipping emissions on O<sub>3</sub> concentrations across China during the period from 2016 to 2020, and explore mechanisms by which shipping emissions influence O<sub>3</sub> formation. Results indicate that shipping emissions increase O<sub>3</sub> concentrations by a five-year average of 3.5 ppb nationwide, exhibiting significant spatial and temporal heterogeneity across different regions and seasons. Although significant differences exist between the emissions of ocean vessels and inland vessels, their contributions to O<sub>3</sub> formation are becoming increasingly comparable. Solely controlling shipping emissions may has limited impact on O<sub>3</sub> mitigation. Instead, coordinated reductions targeting both shipping and land-based anthropogenic sources, along with regionspecific and targeted emission control strategies, are critical for achieving substantial improvements in O<sub>3</sub> pollution mitigation.

29

28

#### 1 Introduction

Over the past decades, China's rapid industrialization and urbanization have boost the economy but also exacerbated air pollution (Zhang et al., 2019). To address air pollution and the health burden from anthropogenic emissions, China introduced the Air Pollution Prevention and Control Action Plan in 2013 (Zheng et al., 2018). Although China has implemented synergistic control of VOC and NO<sub>x</sub> emissions, the warm-season mean maximum daily 8 h average ozone (MDA8 O<sub>3</sub>) increased by 2.6 µg m<sup>-3</sup> yr<sup>-1</sup> in China between 2013 and 2020, especially in urban areas where declining PM<sub>2.5</sub> levels offset gains in O<sub>3</sub> mitigation (Liu et al., 2023). Numerous epidemiological studies have shown that ground-level O<sub>3</sub> pollution leads to a range of adverse health effects, including increased incidence and mortality of respiratory diseases (Ito et al., 2005; Jerrett et al., 2009; Tao et al., 2012). Therefore, the health benefits achieved by reducing PM<sub>2.5</sub> pollution are partially offset by the increase in O<sub>3</sub> pollution (Wang et al., 2021a; Xie et al., 2019), indicating a need to explore strategies to mitigate O<sub>3</sub> pollution in China.

During the promotion of China's emission control actions, emissions from the industry and power sectors declined substantially, with  $NO_x$  reductions exceeding 50%, while the transportation sector still retains significant potential for further cuts (Liu et al., 2023). Ships emit both gaseous and particulate pollutants, including sulfur dioxide ( $SO_2$ ), nitrogen oxides ( $NO_x$ ), particulate matter, and volatile organic compounds (VOC). As the largest maritime trading nation, China has a higher share of shipping emissions among anthropogenic sources compared to international levels (Fu et al., 2017; Yi et al., 2025). From 2016 to 2019, shipping emission controls in China focused on reducing  $SO_2$  and PM emissions through the adoption of low-sulfur fuels, while  $NO_x$  and VOC emissions from shipping continued to rise by approximately 13% due to increasing trade volumes (Wang et al., 2021b). Additionally, the use of low-sulfur fuel may further increase VOC emissions (Wu et al., 2020). Therefore, clarifying the historical and current contribution of shipping emissions to the formation of  $O_3$  is critically important for further pollution control in China.

Previous studies have quantified the impacts of shipping emissions on O<sub>3</sub> pollution in China. In the southern coastal region, shipping emissions contributed approximately 0.9 μg/m<sup>3</sup> to annual O<sub>3</sub> pollution (Cheng et al., 2023), with a peak winter contribution of up to 10% (Feng et al., 2023). In the eastern coastal region during summer, the shipping-related O<sub>3</sub> concentration ranged from -15 to 15 ppb (Fu et al., 2023; Wang et al., 2019a). In the Bohai Rim Area (BRA), shipping emissions showed a maximum annual negative contribution of 0.5 μg/m<sup>3</sup> (Wan et al., 2023), while summer O<sub>3</sub> concentration in Shandong Province increased by up to 10 ppb due to shipping (Wang et al., 2022).. However, these studies predominantly focused on coastal areas, such as Yangtze River Delta (YRD), Pearl River Delta (PRD), and BRA, while the potential inland impacts are not well studied. However, recent studies have indicated that the air quality effects of inland shipping should not be neglected (Huang et al., 2022; Luo et al., 2024). The formation of O<sub>3</sub> exhibits strong spatial heterogeneity and is influenced by multiple factors, such as meteorological conditions and the emission intensities from both shipping and land-based sources. Currently, there is a lack of comprehensive studies employing a unified methodology to assess the nationwide impact of shipping emissions on O<sub>3</sub>. Furthermore, previous studies

were limited to restricted periods, resulting in a deficiency in comprehensive assessments of multi-year scenarios that encompass coordinated variations in both land and shipping emissions.

O<sub>3</sub> is generated by photochemical reactions between NO<sub>x</sub> and VOC under solar radiation, thus, the impacts of shipping emissions on O<sub>3</sub> concentrations are attributed by the nonlinear response of O<sub>3</sub> to changes in NO<sub>x</sub> and VOC emissions (Wang et al., 2019a, 2017). Additionally, the titration of O<sub>3</sub> by NO from shipping emissions, particularly within a few kilometers of ship tracks, can further complicate the simulation and interpretation of O<sub>3</sub> concentrations at the local scale (Merico et al., 2016). However, previous studies commonly used the zero-out method to assess ship's impacts by comparing scenario differences simulated by chemical transport models, which does not fully involve the nonlinear response of O<sub>3</sub> to its precursors and would result in considerable uncertainty in the evaluations (Cheng et al., 2023; Feng et al., 2023; Fu et al., 2023; Wan et al., 2023; Wang et al., 2022, 2019a). Furthermore, although model-based assessments can generate large amounts of simulation data to investigate the impacts of shipping emissions, the number of scenarios that can be simulated by chemical transport models remains limited due to computational constraints. As a result, current analyses struggle to struggles the mechanism of how shipping emissions contribute to O<sub>3</sub> formation from these discrete scenarios. Recently, the advancement of machine learning techniques, with strong capabilities in capturing nonlinear relationships, provides a valuable approach for uncovering underlying patterns in such datasets (Luo et al., 2025).

In this study, we conducted source-oriented chemical transport model with a spatial resolution of 36 km×36 km, to investigate the annual and seasonal impacts of shipping emissions on O<sub>3</sub> concentrations in China, especially for key coastal and inland regions from 2016 to 2020. We also apportion the contribution of shipping emissions from ocean-going vessels (OGVs), coastal vessels (CVs), and river vessels (RVs) to O<sub>3</sub> pollution to identify the influences of regionally differentiated shipping emission control policies. Furthermore, an explainable machine learning model was applied to explore investigate the potential source-receptor relationships between shipping emissions and the O<sub>3</sub> formation based on five-years simulated data. Our study provides a nationwide and long-term analysis of the impacts shipping emissions on China's O<sub>3</sub> pollution, and provide new insights for shipping control measures in the future.

## 2 Methods

## 2.1 Shipping emissions

The Shipping Emission Inventory Model (SEIM v2.0) is a disaggregate dynamic method (Wang et al., 2021b) driven by (a) the high-frequency ship Automatic Identification System (AIS) data, including signal time, coordinate location, navigational speed, and operating status, and (b) the integrated Ship Technical Specifications Database (STSD) (updated to 2020), which describes ship static properties, including vessel type, maximum designed speed, DWT and engine power. First, the originally collected raw AIS data and ship profile data from multiple sources are combined to form a ship activity database and STSD. Second, a route restoration module is applied for cross-land trajectory with a long distance in the AIS data, in which the 10 min linear interpolation will be applied on the shorted paths instead. Third, the

instantaneous emission along with the movement of the ship's trajectory will be calculated based on the ship's static technical parameters, dynamic load changes, and extra parameters and factors. In the SEIM, shipping emissions for both air pollutants (e.g., SO<sub>2</sub>, PM, NO<sub>x</sub>, CO and VOC) and greenhouse gases (e.g., CO<sub>2</sub>, CH<sub>4</sub> and N<sub>2</sub>O) from the main engines, auxiliary engines and boilers were calculated, detailed information of SEIM is described in our previous study (Wang et al., 2021b). Here, emissions beyond 200 nautical miles from the Chinese mainland's territorial sea baseline were excluded from the domain by applying GIS-based spatial processing to the global shipping emission inventory, and only the annual shipping emissions from 2016 to 2020 within 200 nautical miles were used in the simulation.

In this study, vessels were classified as ocean-going vessels (OGVs), coastal vessels (CVs) and river vessels (RVs) for emission estimation according the following rules: (a) OGVs were identified by both valid International Maritime Organization (IMO) numbers and the Maritime Mobile Service Identity (MMSI) numbers, since they are mostly engaged in international trade following the management of the IMO; (b) RVs were identified by frequency distribution method based on the navigation trajectories for each vessel. Vessels with more than 50 % of the AIS signals throughout the entire year occurring on inland rivers (14–43° N, 104–130° E) were considered as RVs; and (c) vessels that are not identified as OGVs or RVs are regarded as CVs. Figure 1 shows the interannual variation of shipping NO<sub>x</sub> and VOC emissions from 2016 to 2020. Overall, the growing trade demands and total cargo throughput of Chinese ports contributed to increased shipping activities, which in turn resulted in a steady rise in shipping NO<sub>x</sub> and VOC emissions, especially for OGVs. Additionally, following the implementation of the global sulfur cap (IMO, 2018), the shift to low-sulfur fuels, which are typically richer in short-chain hydrocarbons (Wu et al., 2020), has contributed to a rise in shipping VOC emissions. It is worth noting that changes in vessel operating conditions, such as idling time and engine load, also influenced emissions. Although the COVID-19 pandemic had a temporary effect on maritime activity, its impact on annual shipping emissions was relatively minor due to the rapid rebound in trade during the second half of the year (Yi et al., 2024).

5

1112

1920

2122

2324

25

26

27

**Figure 1.** The interannual variation of shipping (a)  $NO_x$  and (b) VOC emissions from 2016 to 2020.

## 2.2 Air quality model

 The Weather Research and Forecasting (WRF, version 3.8.1, using meteorological fields from 2018, as detailed later)—Community Multiscale Air Quality (CMAQ, version 5.4) model was applied to simulate the air quality in China from 2016 to 2020. Considering the relatively stable monthly anthropogenic emissions, this study simulated the O<sub>3</sub> concentrations during January, April, July, and November to represent winter, spring, summer, and fall, respectively, for the calculation of annual and seasonal mean values. As shown in **Figure S2**, the modeling domain covered all of China and some parts of East Asia with a horizontal resolution of 36 km × 36 km. Here, we have defined four key regions: BRA, YRD, PRD and inland river areas (Moirangthem, 2016), in which we focus on shipping-related O<sub>3</sub> pollution.

Here, we primarily focused on examining the impact of anthropogenic emission changes on shipping-related O<sub>3</sub> from a historical perspective. To eliminate the impact of interannual meteorological variability, we used meteorological field of 2018 (Zhao et al., 2022), which simulated by WRF and identified as a typical meteorological year due to its relatively stable climate conditions, to drive the CMAQ simulations for the period 2016-2020. The first guess field and boundary conditions for WRF were generated from the 6 h NCEP FNL Operational Model Global Tropospheric Analyses dataset. The four-dimensional data assimilation (FDDA) was enabled using the NCEP ADP global surface and upper air observational weather data (http://rda.ucar.edu, last access: 25 March 2023). WRF and CMAQ used 32 vertical layers up to 100 hPa, and the lowest layer had a thickness of approximately 37 m. The major physical options in WRF included a Morrison two-moment microphysics scheme (Morrison et al., 2009), a Kain-Fritsch cumulus cloud parameterization (Kain, 2004), the Rapid Radiative Transfer Model (RRTM) longwave and shortwave radiation scheme (Iacono et al., 2008), the Pleim-Xiu Land Surface Model (Xiu and Pleim, 2001), and the Asymmetric Convective Model version 3.0 for the PBL parameterization (Pleim, 2007). The distribution of meteorological stations for validation and WRF performance is shown in Figure S3 and Table S1, respectively.

Atmospheric gas-phase chemistry in the CMAQ was simulated with the SAPRC07tic chemical mechanism, and aerosols were predicted using the AERO7. The chemical boundary conditions of CMAQ inputs, corresponding to each simulation period, were collected from the Community Atmosphere Model with Chemistry (CAM-chem) simulation output of global tropospheric and stratospheric compositions (Buchholz et al., 2019). Each run included a 3-day spin-up period. In this study, the Integrated Source Apportionment Method (ISAM) was applied to determine the source contribution to the ambient O<sub>3</sub> concentrations. Here, ISAM-OP3 was applied to attribute all secondary products to sources emitting NO<sub>x</sub> or reactive VOC species and radicals when present in the parent reactants, and otherwise assign them based on stoichiometric reaction rates (Shu et al., 2023). We divided the emissions into five groups to trace them separately in the ISAM, including the land-based anthropogenic emission (the mobiles, industry, power, domestic, and agriculture) from the MEIC and the open burning emissions from Cai's study (Cai et al., 2017), the RVs' emission, the CVs' emission, the OGVs' emission, and the other emission (the nature sources emission from the MEGANv3 and the anthropogenic

emission from other countries within the modeling domain from the MIX (Li et al., 2017), details of emissions are shown in **Table S2**.

We evaluated the simulated MDA8  $O_3$  concentrations for the of 2018 against 1455 available ground-based observations (**Figure S3**) for model validation. As shown in **Table S3**, the simulated MDA8  $O_3$  agreed well with observations, with the overall model performance within the performance criteria suggested by Boylan and Russell (Boylan and Russell, 2006) (mean fractional bias (MFB)  $\leq \pm 60$  % and mean fractional error (MFE)  $\leq \pm 75$  %), while the model overestimated  $O_3$  a little, mainly due to uncertainties in emission inventory and unavoidable deficiencies during meteorological and air quality simulation. Meteorological performance for simulated periods was described in our previous study (Zhao et al., 2022).

## 2.3 Explainable machine learning model

Although CMAQ-ISAM can generate large amounts of simulation data to investigate the impacts of shipping emissions, the number of scenarios remains limited due to computational constraints. As a result, current analyses struggle to elucidate the mechanisms by which shipping emissions contribute to O<sub>3</sub> formation from these discrete scenarios. In particular, capturing nonlinear interactions between emission sources, meteorological conditions, and chemical processes is challenging when only a limited number of emission perturbations are available. Recently, the advancement of machine learning techniques, especially explainable models, has provided a promising complementary approach (Liu et al., 2025; Yao et al., 2024a, b). These models can learn from existing models to approximate the source-receptor relationships embedded in the simulation results. By identifying key emission drivers, quantifying their nonlinear contributions to O<sub>3</sub>, and revealing latent patterns across spatiotemporal scales.

Here, based on the simulated data from 2016 to 2020 using WRF-CMAQ-ISAM, the RF model was used to simulated the monthly average O<sub>3</sub> concentration. In the RF model, the input predictor variables included relative humidity, temperature, wind speed, wind direction, solar radiation, land anthropogenic NO<sub>x</sub> emissions, land anthropogenic VOC emissions, shipping NO<sub>x</sub> emission and shipping VOC emission. Notably, the emissions selected were the sum of emissions from each grid and its eight neighboring grids. We trained four RF models specifically for the BRA, YRD, PRD, and IRA regions, respectively. The simulation data from 2016 to 2019 were used as the training samples, while the simulation data from 2020, under the scenario of the most significant changes in shipping emissions, used as the test samples to validate the generalization capability of the RF models. By comparing the root mean squared error (RMSE) for testing datasets across models with candidate parameter combinations, we set mtry and NumTrees as 6 and 200 in RF, respectively. Additionally, the 10-fold crossvalidation repeated 10 times was considered to evaluate the prediction performance of our models. The total dataset was randomly divided into 10 subsets, where 9 subsets was used to train the model and another was applied for validation. As shown in Figure 2. averages of RMSE and correlation coefficient (R2) in the CV of the RF models were 2.12~2.47 ppb, and 0.90~0.98, respectively, indicating an acceptable performance.

**Figure 2.** Performances of RF models for (a) BRA, (b) YRD, (c) PRD and (d) IRA. Each point represents the monthly average O<sub>3</sub> concentration at each CMAQ grid cell.

In order to identify the sensitivity and response relationship between prediction variables and results in the RF models, the SHapley Additive exPlanations (SHAP) technique, a gametheoretic framework introduced by Lundberg et al. (Lundberg et al., 2020; Lundberg and Lee, 2017), was employed to interpret the pattern learned from the 2016-2019 simulation data by the RF model using the Python scikit-learn library. The SHAP approach enables the quantification of both global and local influences of input variables on the model's predictions, thereby improving the interpretability of factors contributing to air pollution. Additionally, the study examined feature interactions, which can affect the model's predictive accuracy, to gain a more comprehensive understanding of the intricate relationships among the variables.

## 2.4 Limitations

In this study, the spatial resolution of  $36 \text{ km} \times 36 \text{ km}$  may not fully capture the fine-scale spatial heterogeneity of  $O_3$  concentrations, particularly in coastal urban areas where emissions and photochemical reactions exhibit strong spatial variability. This resolution is relatively coarse for accurately representing  $O_3$  exceedances and local photochemical processes, which often occur at much finer spatial scales. Consequently, localized  $O_3$  peaks and gradients may be underestimated or smoothed in the model outputs. Despite this limitation, the selected resolution represents a practical compromise that enables multi-year simulations across the national domain.

Only four representative months were simulated each year to reflect annual and seasonal patterns. While this captures broad seasonal variability, it may overlook intra-seasonal fluctuations and short-term anomalies. Using these months to estimate annual and seasonal means introduces uncertainty, especially for sources with stronger monthly variation. Although monthly changes in anthropogenic and shipping emissions are generally modest (except in 2020), future work could benefit from higher temporal resolution to improve accuracy.

Anthropogenic emissions from other countries within the modeling domain were held fixed at 2010 levels, and open burning emissions were fixed at 2015 levels throughout the simulation period (2016–2020). Although this assumption simplifies the modeling framework and is unlikely to significantly alter the relative changes in shipping-related O<sub>3</sub> assessed in this work, it may still introduce some degree of uncertainty, particularly in regions where long-range transport or fire-related emissions could have contributed more dynamically during specific years. Future studies could benefit from incorporating temporally varying background emissions to further reduce potential uncertainties and improve the representation of external influences.

Explainable machine learning model relies on the structure and quality of the input dataset and cannot account for unmeasured or omitted variables, such as hemispheric background O<sub>3</sub> concentrations. As a result, the derived feature importance reflects statistical associations rather than causal relationships. It should be noted that if one seeks to determine whether a given variable promotes or suppresses O<sub>3</sub> pollution using machine learning methods, additional field observations, experimental data, and corresponding simulation results may be required as supporting evidence. Considering the interactions among variables, even if individual contributions are small, the SHAP estimates for each explanatory variable are unlikely to perfectly reflect their actual contributions in the underlying physical processes. Furthermore, in the presence of strong collinearity or complex nonlinear interactions, SHAP values may not fully disentangle overlapping influences among features.

In this study, monthly and annual mean O<sub>3</sub> concentrations were derived from hourly model outputs, rather than the widely used MDA8 O<sub>3</sub>. While this approach is consistent with the study's focus on long-term trends and average responses, it may introduce bias due to the well-known overestimation of nighttime ozone in chemical transport models. A sensitivity test comparing shipping-related O<sub>3</sub> contributions based on hourly averages and MDA8 revealed that over oceanic areas, the difference may reach 2-5 ppb, while over land, it remains within 2 ppb. Given that the relative contribution of shipping emissions to total O<sub>3</sub> is generally low, the impact of this bias is expected to be limited.

## 3 Results and discussions

#### 3.1 Annual O<sub>3</sub> impact from shipping emissions

**Figure 3a** shows the five-year average of shipping-related O<sub>3</sub> calculated based on hourly values, which is defined as the sum of O<sub>3</sub> concentration caused by emissions of OGVs, CVs, and RVs traced by CMAQ-ISAM. Overall, the shipping emissions increases O<sub>3</sub> concentrations by 3.5 ppb nationwide (**Table S4**), showing a decrease trend from the southeast coast toward inland areas. This result is greater than the findings in other countries such as 1.97 ppb in Europe, and 2.08 ppb in the United States under ambient temperature and pressure (Sun et al., 2024). Due to the high coastal

shipping emissions, the shipping-related O<sub>3</sub> could exceed 15.0 ppb in southeast coastal regions, especially in YRD and PRD, where maximum values reach 25.4 ppb and 26.3 ppb, respectively. For the regions with low shipping emissions, the shipping-related O<sub>3</sub> is relatively low, not exceeding 5 ppb. For example, in IRA and BRA, the shipping-related O<sub>3</sub> is 3.9 ppb and 4.5 ppb respectively, but is slightly higher than the national average. Meteorological factors are just as important as anthropogenic emission influences in O<sub>3</sub> production (Liu et al., 2023; Zhang et al., 2024). For example, the PRD region is characterized by a persistently warm and humid climate, which provides favorable conditions for ozone formation. In contrast, although the BRA is also a coastal region, it experiences lower temperatures and weaker solar radiation (Figure S4), which reduces the formation of hydroxyl radicals, weaking the atmospheric photochemical oxidizing capacity and ultimately limiting the O<sub>3</sub> production.

**Figure 3b** shows the five-year average of the relative contribution of shipping emissions to O<sub>3</sub>. Nationwide, the shipping emissions accounts for a 8.6% increase in O<sub>3</sub>, showing a similar decreasing trend of from the southeast coast to inland regions. This result is higher than 3.7% reported for the Mediterranean region in 2015 (Fink et al., 2023), but lower than the 12-21% reported in another European study for 2010 (Lupascu and Butler, 2019). Notably, some coastal cities exhibit particularly high values. For example, in PRD region, such as in Shenzhen, Guangdong Province, the relative shipping-related O<sub>3</sub> exceeds 30.4%, due to the high intensity of shipping emissions combined with relatively low emissions from land-based anthropogenic sources such as industry and power generation. Similarly, the IRA is relatively low at 10.4%, mainly because of the higher background O<sub>3</sub> concentrations from land-based anthropogenic sources.

**Figure 3.** Contributions of shipping emissions to the five-year average O<sub>3</sub> pollution, including (a) absolute contributions and (b) relative contributions. Maps created with MeteoInfoMap (http://www.meteothink.org).

**Figure 4** illustrates the interannual trend in shipping-related  $O_3$  in key regions from 2016 to 2020. Nationwide, the shipping-related  $O_3$  shows a slight upward trend, with an average annual growth rate of 1.7%, primarily observed in coastal regions. This trend aligns with the changes in shipping  $NO_x$  and VOC emissions, especially in 2020 when a 0.2-0.3 ppb rise in shipping-related  $O_3$  was observed, partly attributable to the notable increase in VOC emissions following the implementation of the global sulfur cap. However, this increase in  $O_3$  concentrations is substantially

lower than the growth in precursor emissions, highlighting the complex and nonlinear response of  $O_3$  formation to changes in shipping emissions. This is because the formation of  $O_3$  depends on photochemical reactions involving  $NO_x$  and VOC under solar radiation, and is influenced not only by the level of shipping emissions but also by land-based anthropogenic emissions, meteorological conditions, and long-range transport (Ye et al., 2023). Therefore, changes in shipping-related  $O_3$  do not scale linearly with the changes in shipping  $NO_x$  and VOC emissions. The Chinese government's two phases of clean air actions (Phase I, 2013–2017; Phase II, 2018–2020) resulted in increasing trend of  $O_3$  nationwide (Liu et al., 2023), and the relative contribution of shipping emissions to  $O_3$  also rose slightly during the same period. It is worth noting that, despite continuous increases in shipping  $NO_x$  and VOC emissions, their relative contributions to  $O_3$  decreased in 2018 and 2020. This pattern may result from simultaneous land-based emission reductions, which can affect atmospheric oxidizing capacity (Lv et al., 2020).

**Figure 4.** The interannual trend in shipping-related O<sub>3</sub>, including (a) absolute contributions and (b) relative contributions, in key regions from 2016 to 2020.

#### 3.2 Contribution of different types of vessels

We further investigated the relationship between O<sub>3</sub> pollution and shipping emissions from sub-ship sectors. **Figure 5** and **Table S4** shows the five-year average contribution of emissions from different ship types to the shipping-related O<sub>3</sub> and the total O<sub>3</sub>, respectively. Nationwide, OGVs, CVs, and RVs contributed 2.6%, 2.6%, and 3.3% to the total O<sub>3</sub>, respectively. In coastal regions, OGVs contribute more than 50% to the shipping-related O<sub>3</sub>, and 9.7% to the total O<sub>3</sub>. CVs are the second-largest contributor to O<sub>3</sub> pollution with an average contribution of 20-30%, and contribute up to 40% in the southern coastal regions near Zhejiang and Fujian Provinces. Notably, RVs can still contribute over 30% to shipping-related O<sub>3</sub> pollution in some coastal regions, particularly in the YRD and PRD. This is primarily due to the presence of major inland waterways such as the lower Yangtze River and the Pearl River, respectively, as well as the influence of regional pollutant transport that enhances the impact of RVs' emissions in these areas. In inland regions, RVs remain the main source of shipping-related O<sub>3</sub>, with contributions in the middle reaches of the Yangtze River reaching 50%.

**Figure 5**. Contributions of emissions from (a) OGVs, (b) CVs, and (c) RVs to the five-year average shipping-related O<sub>3</sub> pollution. Maps created with MeteoInfoMap (http://www.meteothink.org).

The interannual variation in the contribution of the different types of ship to shipping-related O<sub>3</sub> between 2016 and 2020 for different regions is illustrated in **Figure 6**. Nationwide, the contribution of OGVs and RVs increased by 2.7% and 0.6%, respectively, while the contribution of CVs decreased by 3.3%. This pattern was observed in all coastal regions and IRA. The interannual variation in the contribution of the different types of ship to shipping-related O<sub>3</sub> follows a similar pattern to that of shipping-related NO<sub>x</sub> and VOC (**Figure 1**), particularly NO<sub>x</sub> emissions, which shows an upward trend for OGVs and RVs but a downward trend for CVs. As a result, the difference in the contribution of different types of ships to air quality is gradually narrowing. Notably, although RVs emissions significantly less than those of OGVs and CVs, its contribution to O<sub>3</sub> is comparable to that of other ship types, even exceeds that of CVs in some coastal regions. In addition, although China has required certain categories of ships to install AIS equipment since 2010, a large part of small RVs in China have not been equipped with AIS (Zhang et al., 2017), which is not considered in this study. This result suggests the importance of paying greater attention to RVs in future emission control strategies.

**Figure 6.** The interannual trend in Contributions of emissions from OGVs, CVs, and RVs to shipping-related O<sub>3</sub> in key regions from 2016 to 2020.

## 3.3 Seasonal O<sub>3</sub> impact from shipping emissions

The five-years-average seasonal variations in the contribution of shipping emissions to O<sub>3</sub> concentrations across different regions are shown in Figure 7 and Tabel 1, with January, April, July, and November representing winter, spring, summer, and fall, respectively. For cold seasons, including winter and fall, due to weaker solar radiation and lower temperatures that limit O<sub>3</sub> formation (Figure S4), the shipping-related O<sub>3</sub> remains relatively lower than warm seasons (spring and summer), with national average and relative contribution of 1.53 ppb (5.6%) and 2.41 ppb (7.9%), respectively (Figure S5). However, in the south of PRD, especially Guangdong and Hainan Provinces (Figure 7a, 7d, and Table 1), the average and maximum of seasonal shipping-related O<sub>3</sub> exceeds 5 ppb and 21 ppb, respectively, Notably, fall pollution even severer than that in summer. This is mainly because the PRD remains warm and humid in fall, and prevailing monsoon winds are more likely to transport ship-borne pollutants from the sea to inland areas (Figure S4, S6, and S7). Another distinct pattern is observed in BRA, where shipping-related O<sub>3</sub> formation tends to be more localized during the cold seasons, as indicated by a larger difference between the median and average values (Table 1). During this period, mainland China is under the influence of the Mongolian High Pressure System, and continental winds generally suppress the inland transport of ship-related O<sub>3</sub> (Cheng et al., 2023; Zhao et al., 2023). Therefore, significant shipping-related O<sub>3</sub> pollution only appears in major port cities with intensive maritime activity.

In spring, shipping-related O<sub>3</sub> reached its peak in YRD and PRD, with the maximum value exceeding 30 ppb (**Figure 7b** and **Table 1**), consistent with the results of previous studies (Cheng et al., 2023; Schwarzkopf et al., 2022). Although spring is generally less favorable for O<sub>3</sub> formation compared to summer in terms of temperature and humidity, strong onshore winds may play an important role in reduce the influence of shipping emissions (Cheng et al., 2023; Ma et al., 2022) (**Figure S4**, **S6**, and **S7**). In addition, more complex physicochemical interactions may drive springtime O<sub>3</sub> (Cao et al., 2024; Zhang et al., 2024), which needs further investigation. In summer, shipping emissions significantly increased O<sub>3</sub> concentrations nationwide by 4.77 ppb and responsible for 13.7% of national O<sub>3</sub> pollution (**Figure 7c** and **Figure S5**). Notably, even in IRA, where shipping emissions are much lower than in coastal regions, shipping-related O<sub>3</sub> were comparable to those along the coast. This is primarily because central China lies in a perennial monsoon region, where summer monsoons can carry shipping-related air pollutants inland from coastal cities (Zheng et al., 2024).

These findings indicate that seasonal variations for shipping-related O<sub>3</sub> are driven by meteorological factors, particularly changes in prevailing wind direction, which are crucial for the diffusion and long-range transport of shipping emissions. Researchers suggest that the mixing emissions between shipping and local anthropogenic emissions can amplify complicated O<sub>3</sub> chemical formation in coastal cities (Wang et al., 2019b). Therefore, seasonal mitigation strategies and a better understanding of regional monsoon dynamics and their interaction with local anthropogenic emissions are crucial for effectively reducing shipping-related O<sub>3</sub> pollution.

**Figure 7.** Contributions of shipping emissions to the seasonal mean O<sub>3</sub> concentrations for (a) winter (JAN), (b) spring (APR), (c) summer (JUL), and (d) fall (NOV). Maps created with MeteoInfoMap (<a href="http://www.meteothink.org">http://www.meteothink.org</a>).

Table 1 Seasonal ranges of shipping-related O<sub>3</sub> across BRA, YRD, PRD, and IRA. (unit: ppb)

| Region | Metric       | Winter | Spring | Summer | Fall  |
|--------|--------------|--------|--------|--------|-------|
| BRA    | Minimum      | 0.01   | 0.83   | 1.25   | 0.06  |
|        | 25% Quartile | 0.05   | 3.09   | 7.13   | 0.39  |
|        | Median       | 0.11   | 4.02   | 9.32   | 0.78  |
|        | 75% Quartile | 0.91   | 6.55   | 11.90  | 1.63  |
|        | Maximum      | 6.59   | 23.22  | 32.40  | 14.39 |
|        | Mean         | 0.71   | 5.36   | 10.13  | 1.74  |
| YRD    | Minimum      | 0.20   | 1.92   | 1.45   | 0.55  |
|        | 25% Quartile | 1.01   | 5.02   | 6.11   | 1.91  |
|        | Median       | 1.79   | 6.64   | 7.29   | 3.23  |
|        | 75% Quartile | 2.93   | 8.35   | 9.03   | 5.14  |

|     | Maximum      | 16.32 | 31.47 | 25.86 | 24.59 |
|-----|--------------|-------|-------|-------|-------|
|     | Mean         | 2.47  | 7.39  | 7.92  | 4.41  |
| PRD | Minimum      | 1.11  | 3.91  | 0.11  | 1.91  |
|     | 25% Quartile | 2.79  | 7.94  | 3.19  | 4.85  |
|     | Median       | 5.19  | 10.07 | 5.65  | 7.97  |
|     | 75% Quartile | 7.68  | 15.17 | 7.77  | 11.25 |
|     | Maximum      | 21.98 | 33.46 | 26.19 | 28.78 |
|     | Mean         | 5.96  | 11.91 | 5.77  | 8.79  |
| IRA | Minimum      | 0.17  | 0.91  | 2.32  | 0.10  |
|     | 25% Quartile | 1.19  | 2.88  | 5.65  | 1.44  |
|     | Median       | 1.61  | 4.85  | 9.07  | 2.85  |
|     | 75% Quartile | 2.29  | 5.73  | 11.31 | 4.16  |
|     | Maximum      | 5.53  | 11.91 | 26.19 | 7.52  |
|     | Mean         | 1.76  | 4.58  | 9.03  | 2.87  |

## 3.4 Effects of shipping emission on O<sub>3</sub> formation

Although the features in the RF model include both meteorological factors and emissions, this section focuses on impacts of anthropogenic emissions, especially shipping emissions, on O<sub>3</sub> pollution. **Figure 8** presents the SHAP summary plots for selected features in the RF model for BRA, YRD, PRD, and IRA, showing the magnitude, prevalence, and direction of each feature's impact on the model output (O<sub>3</sub> concentration). In the summary plot, the further a feature's SHAP value is from zero, the greater its influence; positive SHAP values indicate a positive contribution, while negative values indicate a negative effect. For example, in **Figure 8a**, land-based NO<sub>x</sub> and VOC emissions both have a significant impact on O<sub>3</sub> formation in the BRA, with contributions of 16.8% and 11.0%, respectively, suggesting that the atmospheric chemistry in this area is significantly affected by land-based anthropogenic emissions. Moreover, lower NO<sub>x</sub> and VOC emissions leads to higher SHAP values, indicating a negative correlation between land-based anthropogenic emissions and O<sub>3</sub> pollution. In the coastal areas of BRA, shipping emissions are much smaller than land-based emissions, therefore, contribute only approximately 2.9% to O<sub>3</sub> formation, and exhibit a similar negative correlation as land-based sources.

As the share of shipping emissions increases within total anthropogenic emissions in the coastal areas of YRD and PRD, the difference between the contributions of shipping and land-based emissions to  $O_3$  pollution regions decreases. Especially in the PRD, shipping  $NO_x$ 

contributes up to 9.7%, exceeding the land-based VOC contribution of 5.3%. In the IRA, emissions from inland vessels are much lower than those from ocean-going and coastal ships, thus, shipping NO and VOC contributions only 1.5% and 0.8% to O<sub>3</sub> pollution, respectively.

For meteorological factors, the contributions of solar radiation (25.2%), temperature (20.0%), wind speed (11.5%), and wind direction (4.9%), relative humidity (4.8%) to O<sub>3</sub> formation exhibit clear regional heterogeneity. Overall, meteorological influences are greater than those of shipping emissions. This may be attributed to the highly complex physical and chemical processes involved, including cloud–radiation interactions, air mass transport, and water-related reactions.

**Figure 8.** Feature importance results of the random forest regression model for (a) BRA, (b) YRD, (c) PRD, and (d) IRA. The x-axis shows SHAP values representing the impact of each feature on O<sub>3</sub> predictions (positive: increasing O<sub>3</sub>; negative: decreasing O<sub>3</sub>). Each dot is a gridmonth sample, with color indicating the feature value. Instances with identical x-values are stacked, and the stack height signifies the density.

The dependence plot (**Figure. 9**) further quantifies how changes in shipping emissions affect  $O_3$  concentrations. Notably, due to the uneven distribution of shipping emissions, the x-axis in **Figure. 9** is set to non-uniform intervals to better illustrate their impact. Overall, as previously mentioned, changes in shipping  $NO_x$  and VOC emissions have a relatively minor effect on regional  $O_3$  levels—approximately between -2.5 ppb to 8 ppb—although there is clear regional heterogeneity. In the BRA region, shipping emissions are negatively correlated with  $O_3$ , with increases in shipping  $NO_x$  and VOC emissions leading to a reduction of about 1-2 ppb in  $O_3$ . Differently, in the YRD and PRD, increases in shipping  $NO_x$  emissions slightly promote

1

2

4 5

6

7

8

9

12 13

21

O<sub>3</sub> formation-especially in the PRD region, where O<sub>3</sub> may increase by up to 3 ppb, which is opposite to the impact of land-based  $NO_x$  emissions (Figure. S7). This difference may be because, in the YRD and PRD, land-based NOx emissions do not dominate the overall anthropogenic emissions as they do in the BRA, allowing shipping NO<sub>x</sub> to also influence atmospheric chemistry. Furthermore, changes in shipping VOC emissions have almost no impact, consistent with the effect of changes in land-based VOC emissions (Figure. S7). Only when shipping VOC emissions increase dramatically in the YRD region is O<sub>3</sub> formation suppressed—though such cases are rare, as reflected by the partially negative SHAP values for shipping VOC emissions in Figure. 8b. In the IRA, similar to the BRA region, shipping emissions are very small compared to land-based sources, so that changes in shipping NO<sub>x</sub> and VOC emissions have a similar negative effect on O<sub>3</sub> as those from land-based sources.

Figure 9. SHAP dependence plot for (a) BRA, (b) YRD, (c) PRD, and (d) IRA.

## 4 Conclusion

In this study, we conducted multi-year CMAQ-ISAM simulations to investigate the how shipping emissions impacted O<sub>3</sub> across China, with a focus on three coastal regions and a inland region. From 2016 to 2020, shipping emissions increased national average O<sub>3</sub> concentrations by 3.5 ppb, accounting for 8.6% of total O<sub>3</sub>, with a spatial gradient decreasing from coastal to inland regions. Despite the increasing intensity of shipping activity and the implementation of the global sulfur cap, shipping NO<sub>x</sub> and VOCs emissions rose significantly during this period. However, the national average shipping-related O<sub>3</sub> increased by only 0.23 ppb, while the relative contribution of shipping emissions to O<sub>3</sub> pollution rose by approximately 0.5%. Notably, this relative contribution did not increase continuously; instead, a decline was observed in 2018 and 2020. This non-linear response, under conditions of simultaneous changes in multiple pollutants from different sectors, highlights the complexity and need for further investigation of attribution of O<sub>3</sub> pollution. For the four focus regions, the contribution of shipping to O<sub>3</sub> levels exceeded the national average, with more pronounced interannual increases.

We further disaggregate ship types to OGVs, CVs, and RVs. The result revealed that OGVs

were the dominant contributors to shipping-related O<sub>3</sub> in coastal areas, followed by CVs, whereas RVs were the main source in inland river areas. Although OGVs, CVs, and RVs differ significantly in their emission magnitudes, the difference in their contributions to O<sub>3</sub> pollution is gradually narrowing. This trend suggests that the influence of RVs on regional O<sub>3</sub> levels should no longer be overlooked and that emission control efforts for RVs deserve renewed attention. However, from the perspective of sulfur emission control, RVs in China had already reached the final stage of sulfur regulation by 2018 under the implementation of domestic emission control policies. In contrast, NO<sub>x</sub> control for inland vessels remains largely unaddressed. Globally, there is limited precedent or experience in regulating NO<sub>x</sub> emissions from inland waterways, leaving China without a clear reference framework for RVs NO<sub>x</sub> mitigation. Future control of shipping NO<sub>x</sub> emissions needs to take into account both inland waterways and coastal areas.

The impacts of shipping emissions on O<sub>3</sub> also exhibited significant seasonal and regional characteristics. While shipping-related O<sub>3</sub> levels were generally lower in colder seasons, fall pollution in southern coastal regions exceeded that of summer due to favorable land—sea monsoon transport. Peak shipping-related O<sub>3</sub> levels occurred in spring over YRD and PRD, and in summer over inland areas. These patterns highlight the importance of implementing seasonal and region-specific control strategies to mitigate shipping-related O<sub>3</sub> pollution effectively. In particular, quality-oriented management policies such as seasonal routing adjustments, port operation scheduling, or dynamic emission monitoring, may play a more immediate role than emission control policies, which are typically less adaptable to seasonal variability and require long-term infrastructure or regulatory changes. Therefore, combining flexible operational measures with long-term emission reduction plans could enhance the overall effectiveness of O<sub>3</sub> mitigation.

Interpretable machine learning analysis further revealed significant spatial differences in the contribution of shipping emissions to O<sub>3</sub>. In BRA and IRA, O<sub>3</sub> formation was primarily driven by land-based NO<sub>x</sub> and VOC emissions, with shipping emissions playing a minor role and even showing a suppressive effect on O<sub>3</sub> formation. In contrast, in coastal regions such as YRD and PRD, the increasing share of shipping emissions in the total anthropogenic emissions enhanced their contribution to O<sub>3</sub>, with shipping NO<sub>x</sub> emissions showing a slight promoting effect on O<sub>3</sub> formation. This regional difference suggests that solely controlling shipping emissions may lead to unexpected atmospheric chemical responses and, under certain conditions, could even cause an increase in O<sub>3</sub> concentrations. Therefore, effective O<sub>3</sub> pollution control requires a coordinated reduction of both land-based and shipping emissions, based on regional emission structures and atmospheric oxidation characteristics.

## Data availability

Data used during the current study are available from the corresponding author upon reasonable request.

#### Code availability

1

4

- 2 Codes used during the current study are available from the corresponding author upon
- 3 reasonable request.

## Acknowledgments

- This research has been supported by: National Natural Science Foundation of China (grant
- nos. 42325505 to H.L.), National Key Research and Development Program of China (grant no.
- 2022YFC3704200 to H.L.), the China National Postdoctoral Program for Innovative Talents
- (No. BX20250327 to Z.Luo), the Shuimu Tsinghua Scholar Program (no. 2024SM028 to
- Z.Luo), National Natural Science Foundation of China (grant no. 42405097 to Z.Lv).

## **10** Author Contributions

- Conceptualization: Z.Luo, L.P., H.L.; Methodology: Z.Luo, Z.Lv, J.Zhao; Investigation:
- W.Y., T.H., Y.W; Visualization: Z.L.uo L.P.; Supervision: H.L., K.H.; Writing—original draft:
- Z.Luo, L.P.; Writing—review & editing: Z.Luo, H.L.

#### 14 **Declaration of interests**

The authors declare no competing interests.

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

| 1 | emissions since 2010 as the consequence of clean air actions, Atmos. Chem. Phys., 18,              |
|---|----------------------------------------------------------------------------------------------------|
| 2 | 14095-14111, https://doi.org/10.5194/acp-18-14095-2018, 2018.                                      |
| 3 | Zheng, S., Jiang, F., Feng, S., Liu, H., Wang, X., Tian, X., Ying, C., Jia, M., Shen, Y., Lyu, X., |
| 4 | Guo, H., and Cai, Z.: Impact of marine shipping emissions on ozone pollution during the            |
| 5 | warm seasons in China, J. Geophys. Res.: Atmos., 129, e2024JD040864,                               |
| 6 | https://doi.org/10.1029/2024JD040864, 2024.                                                        |
| 7 |                                                                                                    |