# Peer review of "Impacts of shipping emissions on ozone pollution in China"

_EGUsphere, 2025_

## Author Comment (AC1)

**Response to Reviewer #2's Comments**

**General comment**

This manuscript investigates the influence of shipping emissions on surface ozone concentrations in China. To support this analysis, the authors have extended the shipping emission inventory SEIMv2.0 for the year 2020. The study employs the WRF-CMAQ chemical transport model and the ISAM source apportionment module to assess the contributions of ocean-going, coastal, and river vessels to surface ozone concentrations. Additionally, a random forest machine learning model is applied to interpret the sensitivity of monthly mean ozone levels to various input features, including meteorological parameters, land-based anthropogenic emissions, and shipping-related emissions. While the study addresses a timely and important topic, several major concerns should be carefully addressed before the manuscript can be considered for publication.

A primary concern is the limited depth of analysis and clear contribution to current scientific knowledge on ozone pollution. The manuscript uses a style more aligned with a technical report, lacking a thorough link with the current literature on the role of shipping emissions in ozone formation, particularly in coastal and river basin environments. Furthermore, the novelty of the work and its broader implications are not clearly conveyed. A more robust discussion contrasting the study with recent literature studies conducted in other regions would significantly strengthen the manuscript's relevance and potential impact.

Although the paper is generally well structured, several methodological aspects require further clarification. Notably, the stated objective of investigating interannual impacts of shipping emissions from 2016 to 2020 contrasts with the modeling setup, which uses meteorology from a single representative year (2018). This approach primarily assesses the impact of emission changes under fixed meteorological conditions, rather than capturing interannual variability. Emissions from other sources (e.g., international anthropogenic sources from 2010, and open burning from 2015) are also held constant. The implications of this modeling design should be explicitly acknowledged, and the study's objectives reformulated to better reflect the actual scope of the simulations.

In this regards, the presentation of results as a five-year average raises concerns about the interpretation and relevance of the findings. It is unclear what this average represents given the modeling configuration. While averaging can simplify interpretation, it risks obscuring temporal variability and may lead to misleading conclusions about the spatial and seasonal influence of shipping emissions. I strongly recommend avoiding multi-year averaging in this context. Instead, results should be presented as sensitivity simulations under consistent meteorological conditions, with comparisons made between specific emission scenarios (e.g., 2016 vs. 2020). Furthermore, the analysis would benefit greatly from an angle considering high-ozone episodes (e.g., events when MDA8 > 120 μg/m³), as these events are of particular interest for air quality management. Moreover, with a more in-depth statistical analysis, the authors could examine the sensitivity and contributions of the different shipping sources during low-, medium-, and high-ozone concentration events.

The application of an explainable machine learning model to explore ozone drivers is an interesting addition; however, its implementation raises several issues. It is well established that meteorology significantly influences ozone formation, but hemispheric background ozone concentrations also play a crucial role, as highlighted in several recent studies (e.g., Jonson et al., 2018; Lupaşcu and Butler, 2019; Shu et al., 2023; Garatachea et al., 2024). The omission of background ozone as a feature in the machine learning model is a significant limitation and likely biases the interpretation of feature importance. Given that the ISAM module is capable of capturing this background contribution, its integration into the machine learning framework should be considered. Additionally, the use of monthly mean concentrations limits the model's utility for understanding episodic ozone dynamics, which often unfold over shorter timescales. A more granular temporal resolution would be more appropriate for exploring the drivers of ozone exceedances.

The manuscript would benefit from careful proofreading. While the general structure is acceptable, several sections require refinement for clarity and precision.

This manuscript addresses a topic of considerable scientific and policy interest. However, major revisions are required to improve the clarity of the methods, enhance the scientific discussion, and strengthen the novelty and relevance of the findings. I encourage the authors to address the comments above and the specific points provided below before considering the

paper for publication.

**Response:**

Thank you for your valuable feedback. We appreciate the your insights regarding the depth of analysis and clarity of the study's scientific contribution. In response, we have revised the manuscript to enhance the discussion.

We acknowledge the reviewer's concern regarding the consistency between the study's stated objective and the modeling setup. As suggested, we have revised the manuscript to explicitly clarify that our approach focuses on assessing the impact of interannual changes in shipping emissions under fixed meteorological and background emission conditions. This issue has been addressed in detail in our response to Reviewer #1's Comment 6. Correspondingly, we have refined the study objectives to better reflect the actual scope of our simulations. Additionally, the influence of international anthropogenic sources has been discussed in our response to Reviewer #1's Comment 1. Other limitations identified in the modeling design have also been incorporated into a newly added "Limitations" section.

While we acknowledge the potential limitations of multi-year averaging, we would like to clarify that in addition to presenting five-year average spatial patterns, we have also provided interannual variation analyses to support temporal interpretation of our results. The aim of this study is to investigate the long-term impact of shipping emissions on $O_3$ formation under evolving land-based anthropogenic emissions across China, with a focus on identifying trends and providing insights for future shipping emission control strategies. Therefore, our multi-year approach is aligned with this objective. We fully agree that short-term $O_3$ episodes are critical for air quality management; however, such events are not the primary focus of this study. Moreover, given the constraints of the available emission inventories, particularly the lack of high-temporal-resolution emission data, it would be challenging to reliably assess episodic impacts within our current framework. Further discussion on this limitation is provided in our response to Comment 19 and 22.

Regarding the exclusion of background $O_3$ concentrations in the machine learning model, we

believe this does not compromise the validity of the explainable analysis, as detailed in our response to Comment 28. As for the use of monthly mean values, this aligns with the study's objective of exploring long-term trends and average responses, which has been addressed in Comment 19.

Overall, we have substantially revised the manuscript to enhance the clarity and transparency of the methodological framework, refined the expression of key results, and strengthened the scientific discussion throughout. We have deepened the analysis to better highlight the interannual and seasonal characteristics of shipping-related ozone pollution, emphasized the novelty of applying explainable machine learning to regional attribution, and clarified how our findings support differentiated emission control strategies. Additionally, we have aligned the study objectives more closely with the modeling design, and introduced a new "Limitations" section to acknowledge key uncertainties. The manuscript has been thoroughly proofread and edited to improve clarity, precision, and consistency across all sections.

**Revisions in Main Text:**

**Specific comments**

**Comment 1**

Page 1, Line 22: It is unclear what is considered an "effective ozone mitigation measure" in the context of China, and how controlling shipping emissions contributes, for example, to reducing ozone exceedances across the country. This should be clarified in the main text, preferably in the Introduction. A nationwide contribution of 3.5 ppb may be highly relevant if it leads to exceedances in specific regions.

**Response:**

Thank you for the comment. We agree that the term "effective O3 mitigation measure" is vague and can be difficult to define in the context of China, especially given the complex and regionally variable nature of ozone pollution. Additionally, how controlling shipping emissions specifically contributes to reducing nationwide ozone exceedances is indeed a challenging question under current scientific understanding. In response, we have revised the manuscript

to remove or clarify such ambiguous expressions and avoid overstating the implications.

**Revisions in Main Text:**

**1 Introduction**

Solely controlling shipping emissions may has limited impact on $O_3$ mitigation.

**Comment 2**

Page 2, Line 6: Please clarify whether ozone is emerging as a more significant issue in urban or rural regions across China.

**Response:**

Thanks for your comment. We have revised the sentence to make it clearer.

**Revisions in Main Text:**

**1 Introduction**

Although China has implemented synergistic control of VOC and $NO_x$ emissions, the warm-season mean maximum daily 8 h average ozone (MDA8 $O_3$) increased by 2.6 μg m$^{-3}$ yr$^{-1}$ in China between 2013 and 2020, especially in urban areas where declining $PM_{2.5}$ levels offset gains in $O_3$ mitigation (Liu et al., 2023).

**Comment 3**

Page 2, Line 14: A reference should be provided for the reported reductions in ozone and its precursors in China. Is shipping now viewed as a major contributor because emissions from other sectors have already undergone significant reductions?

**Response:**

Thanks for your comment. We have added the relevant references. As for shipping's role, there are currently no studies demonstrating that it is a major contributor to O3 pollution. However, in terms of emissions, reductions from the transportation sector have been much smaller than those from other sectors, and shipping-related NOx emissions have continued to increase. 。

**Revisions in Main Text:**

**1 Introduction**

Although China has implemented synergistic control of VOC and $NO_x$ emissions, the warm-season mean maximum daily 8 h average ozone (MDA8 $O_3$) increased by 2.6 μg m$^{-3}$ yr$^{-1}$ in China between 2013 and 2020, especially in urban areas where declining $PM_{2.5}$ levels offset gains in $O_3$ mitigation (Liu et al., 2023).

During the promotion of China's emission control actions, emissions from the industry and power sectors declined substantially, with $NO_x$ reductions exceeding 50%, while the transportation sector still retains significant potential for further cuts (Liu et al., 2023).

**Comment 4**

Page 2, Line 21: Quantitative estimates of the increase in emissions should be provided.

**Response:**

Done.

**Revisions in Main Text:**

**1 Introduction**

From 2016 to 2019, shipping emission controls in China focused on reducing SO2 and PM emissions through the adoption of low-sulfur fuels, while NOx and VOC emissions from shipping continued to rise by approximately 13% due to increasing trade volumes.

**Comment 5**

Page 2, Line 25: Quantitative results should be presented and compared with findings from similar studies conducted in other regions to support a more comprehensive literature discussion.

**Response:**

Done.

**Revisions in Main Text:**

**1 Introduction**

Previous studies have quantified the impacts of shipping emissions on $O_3$ pollution in China. In the southern coastal region, shipping emissions contributed approximately 0.9 μg/m³ to annual $O_3$ pollution (Cheng et al., 2023), with a peak winter contribution of up to 10% (Feng et al., 2023). In the eastern coastal region during summer, the shipping-related $O_3$ concentration ranged from -15 to 15 ppb (Wang et al., 2019; Fu et al., 2023). In the Bohai Rim Area (BRA), shipping emissions showed a maximum annual negative contribution of 0.5 μg/m³ (Wan et al., 2023), while summer $O_3$ concentration in Shandong Province increased by up to 10 ppb due to shipping (Wang et al., 2022).

**Comment 6**

Page 2, Line 40: Please include relevant references to support the statements made.

**Response:**

Done.

**Revisions in Main Text:**

**1 Introduction**

However, previous studies commonly used the zero-out method to assess ship's impacts by comparing scenario differences simulated by chemical transport models, which does not fully involve the nonlinear response of $O_3$ to its precursors and would result in considerable uncertainty in the evaluations (Wang et al., 2019; Wang et al., 2022; Cheng et al., 2023; Feng et al., 2023; Fu et al., 2023; Wan et al., 2023)..

**Comment 7**

Page 3, Line 6: Is the model resolution employed in this study sufficient to capture ozone exceedances across China? A brief discussion on the setup limitations and the rationale for its selection should be included in the Methods section.

**Response:**

Thanks for your question.

We agree that a finer spatial resolution is generally more appropriate for capturing local-scale

ozone formation processes. However, our objective in this study is to assess the regional and interannual impacts of shipping emissions on ozone pollution at the national scale, rather than focusing on local photochemical processes at the urban or neighborhood level.

Therefore, the selected resolution of 36 km × 36 km represents a practical compromise between spatial detail and computational feasibility, especially considering the need to simulate multi-year scenarios (2016–2020) across the entire Chinese domain. This spatial resolution is also consistent with a series of studies by Geng et al. (as shown in the table below), who have extensively investigated ozone pollution and its driving mechanisms in China using similar model setups. We have added a statement in the Methods section.

| Reference | Model/Spatial resolution |
|---|---|
| Drivers of Increasing Ozone during the Two Phases of Clean Air Actions in China 2013–2020 | WRF-CMAQ/36 km |
| Evaluating the spatiotemporal ozone characteristics with high-resolution predictions in mainland China, 2013–2019 | WRF-CMAQ/36 km |
| Estimating Spatiotemporal Variation in Ambient Ozone Exposure during 2013–2017 Using a Data-Fusion Mode | WRF-CMAQ/36 km |

Additionally, the spatial resolution of the ship emission inventory we constructed is 0.05°, the land-based anthropogenic emission inventory from MEIC has a spatial resolution of 0.25°. Allocating land-based anthropogenic emissions to a much finer grid could significantly increase the uncertainty of the simulation.

**Revisions in Main Text:**

**2.4 Limitations**

In this study, the spatial resolution of 36 km × 36 km may not fully capture the fine-scale spatial heterogeneity of $O_3$ concentrations, particularly in coastal urban areas where emissions and photochemical reactions exhibit strong spatial variability. This resolution is relatively coarse for accurately representing $O_3$ exceedances and local photochemical processes, which often occur at much finer spatial scales. Consequently, localized $O_3$ peaks and gradients may be

underestimated or smoothed in the model outputs. Despite this limitation, the selected resolution represents a practical compromise that enables multi-year simulations across the national domain.

**Comment 8**

Page 3, Line 11: The use of machine learning to analyze the impact of shipping emissions on ozone formation should be better justified, especially considering this is the primary aim of the ISAM source apportionment tool. The added value of the machine learning approach relative to insights already provided by CMAQ-ISAM should be clearly explained.

**Response:**

Thanks for your comment. While CMAQ-ISAM provides an effective means to quantify the contributions of shipping emissions to O3 formation, its application relies on predefined source tagging and discrete simulation scenarios. Exploring the nonlinear and regime-dependent response of O3 to changes in shipping emissions—particularly under varying meteorological and chemical backgrounds—would require a large number of ISAM sensitivity simulations, which are computationally expensive and time-consuming.

To address this limitation, we employed an explainable machine learning model trained on the ISAM-based simulation outputs over five years. This approach allows us to extract and generalize the underlying relationships between shipping precursor emissions ($NO_x$ and VOCs), meteorological conditions, and resulting O3 responses. By doing so, we are able to capture key sensitivities and interactions that are otherwise difficult to obtain through conventional scenario-based modeling alone, and provide interpretable insights that can inform future control strategies.

**Revisions in Main Text:**

**1 Introduction**

Furthermore, although model-based assessments can generate large amounts of simulation data to investigate the impacts of shipping emissions, the number of scenarios that can be simulated by chemical transport models remains limited due to computational constraints. As a result,

current analyses struggle to struggles the mechanism of how shipping emissions contribute to $O_3$ formation from these discrete scenarios. Recently, the advancement of machine learning techniques, with strong capabilities in capturing nonlinear relationships, provides a valuable approach for uncovering underlying patterns in such datasets (Luo et al., 2025).

Furthermore, an explainable machine learning model was applied to explore investigate the potential source-receptor relationships between shipping emissions and the $O_3$ formation based on five-years simulated data.

**Comment 9**

Page 3, Line 28: Please be more specific and avoid overly verbose statements.

**Response:**

Thanks for your comment. We have removed this sentence, as it does not contribute substantially to the understanding of the study.

**Comment 10**

Page 3, Line 32: Clarify whether SEIMv2.0 is extended for 2020 only, and whether a recalculation for the 2016–2019 period was performed. The treatment of emissions beyond 200 nautical miles from the Chinese coastline should also be described. It is currently unclear which emission inventory and reference year are used for those sources (not shown in Table S2).

**Response:**

Thank you for your comment. SEIMv2.0 has been applied for 2016 to 2020, and considering the recent updates in VOC emission factors for low-sulfur fuels, we have also recalculated the emissions for the 2016-2019. We apologize for the previous wrong information in Table S2, which has now been corrected. As for emissions beyond 200 nautical miles from the Chinese coastline, these were extracted from the global shipping emission inventory using Arcgis-based spatial processing based on the corresponding shapefiles, which has now been described.

**Revisions in Main Text:**

**2.1 Shipping emissions**

Here, emissions beyond 200 nautical miles from the Chinese mainland's territorial sea baseline were excluded from the domain by applying GIS-based spatial processing to the global shipping emission inventory, and only the annual shipping emissions from 2016 to 2020 within 200 nautical miles were used in the CMAQ-ISAM simulation.

**Comment 11**

Page 3, Line 36: Define the acronym "IMO" upon first use.

**Response:**

Done.

**Revisions in Main Text:**

**2.1 Shipping emissions**

OGVs were identified by both valid International Maritime Organization (IMO) numbers and the Maritime Mobile Service Identity (MMSI) numbers.

**Comment 12**

Page 4, Line 4: Cite the regulatory measure that enforce the use of low-sulfur fuels in 2020. Additionally, could the observed sudden changes in OGV emissions in 2020 be partly attributed to the COVID-19 pandemic?

**Response:**

Thanks for your comment. Indeed, the COVID-19 pandemic had a short-term impact on global and domestic shipping activities in early 2020. However, with the recovery of maritime trade in the second half of the year, a "rebound effect" in shipping traffic was observed. As a result, the influence of the pandemic on annual shipping emissions in 2020 was relatively limited from an interannual perspective (Yi et al., 2024). We have added a clarification in the revised manuscript to reflect this point.

**Revisions in Main Text:**

**2.1 Shipping emissions**

Additionally, following the implementation of the global sulfur cap (IMO, 2018), the shift to low-sulfur fuels, which are typically richer in short-chain hydrocarbons (Wu et al., 2020), has contributed to a rise in shipping VOC emissions.

Although the COVID-19 pandemic had a temporary effect on maritime activity, its impact on annual shipping emissions was relatively minor due to the rapid rebound in trade during the second half of the year (Yi et al., 2024).

**Comment 13**

Page 4, Line 12: The sentence should be rephrased to clarify that emissions from 2016 to 2020 are simulated using 2018 meteorology. The current wording is misleading. Also, clarify how annual and seasonal means are derived when only 4 out of 12 months are simulated. Justification for this simplification and its limitations should be provided.

**Response:**

Thank you for your comment. We have added a clarification to indicate that the meteorological data used in this study are from the year 2018.

The four selected months-January, April, July, and November-were chosen to represent the four seasons: winter, spring, summer, and fall, respectively, and correspond to meteorologically typical periods throughout the year. For emissions, both shipping and land-based anthropogenic emissions exhibit relatively small month-to-month variability (except for the anomaly observed in 2020 due to the COVID-19 pandemic). Therefore, this seasonal sampling strategy is considered reasonable for capturing the annual and interannual patterns of O3 formation. We have added clarification in the revised manuscript and included a discussion of this simplification and its limitations.

**Revisions in Main Text:**

**2.2 Air quality model**

The Weather Research and Forecasting (WRF, version 3.8.1, using meteorological fields from 2018, as detailed later)−Community Multiscale Air Quality (CMAQ, version 5.4) model was applied to simulate the air quality in China during January, April, July and November from

2016 to 2020.

Considering the relatively stable monthly anthropogenic emissions, this study simulated the $O_3$ concentrations during January, April, July, and November to represent winter, spring, summer, and fall, respectively, for the calculation of annual and seasonal mean values.

**2.4 Limitations**

Only four representative months (January, April, July, and November) were simulated each year to reflect annual and seasonal patterns. While this captures broad seasonal variability, it may overlook intra-seasonal fluctuations and short-term anomalies. Using these months to estimate annual and seasonal means introduces uncertainty, especially for sources with stronger monthly variation. Although monthly changes in anthropogenic and shipping emissions are generally modest (except in 2020), future work could benefit from higher temporal resolution to improve accuracy.

**Comment 14**

Table S1: Include a comprehensive caption that explains how the statistics are computed. Indicate which meteorological stations are used and describe the spatial and temporal aggregation methods.

**Response:**

Done.

**Revisions in Main Text:**

**2.2 Air quality model**

The distribution of meteorological stations for validation and WRF performance is shown in **Figure S3** and Table S1, respectively.

We evaluated the simulated $O_3$ concentrations for the of 2018 against 1455 available ground-based observations (**Figure S3**) for model validation.

**Revisions in Supplement:**

[Figure]

**Figure S3.** Distribution of 401 meteorological (purple dots) and 1455 air pollutant (red dots) observation stations.

* $MB = \frac{\sum_{i=1}^{N} S_i - O_i}{N}$, $NMB = \frac{\sum_{i=1}^{N} S_i - O_i}{\sum_{i=1}^{N} O_i}$, $NMGE = \frac{\sum_{i=1}^{N} |S_i - O_i|}{\sum_{i=1}^{N} O_i}$, $MFB = \frac{2 \times \sum_{i=1}^{N} \frac{S_i - O_i}{S_i + O_i}}{N}$, $MFE = \frac{2 \times \sum_{i=1}^{N} \frac{|S_i - O_i|}{S_i + O_i}}{N}$, $R = \frac{\sum_{i=1}^{N} [(S_i - \bar{S}) \times (O_i - \bar{O})]}{\sqrt{\sum_{i=1}^{N} (S_i - S)^2 \times \sum_{i=1}^{N} (O_i - \bar{O})^2}}$,

* $N$ is the total number of samples in the dataset, $S_i$ is the simulated value of the $i$-th sample, $O_i$ is the observed (measured) value of the $i$-th sample, $\bar{S}$ is the mean of the simulated values, $\bar{O}$ is the mean of the observed values.

**Comment 15**

Page 5, Line 3: Please confirm whether chemical boundary conditions also correspond to the year 2018.

**Response:**

Thanks for your question. The chemical boundary conditions from CAM-chem are not fixed to the year 2018; instead, they correspond to the respective simulation periods from 2016 to 2020, ensuring temporal consistency with the model scenarios.

**Revisions in Main Text:**

**2.2 Air quality model**

The chemical boundary conditions of CMAQ inputs, corresponding to each simulation period, were collected from the Community Atmosphere Model with Chemistry (CAM-chem) simulation output of global tropospheric and stratospheric compositions (Buchholz et al., 2019).

**Comment 16**

Page 5, Line 5: Specify which tagging method within ISAM is used. The authors should justify the choice and discuss its implications, as different tagging schemes may yield significantly different results (see Shu et al., 2023).

**Response:**

Thanks for your comment. We have now specified in the Methods that ISAM-OP3 was used in this study. In preliminary tests we applied ISAM-OP5 over the full 2016–2020 period, but found that it systematically over-attributed O3 to shipping sources. We then ran one-month sensitivity experiments with ISAM-OP1-OP4 and, by comparing our sectoral O3 patterns against existing studies, determined that OP3 produced the most plausible attributions. At present, however, our available simulations and analyses do not support a fully rigorous, mechanistic justification of this choice or its detailed implications.

**Revisions in Main Text:**

**2.2 Air quality model**

Here, ISAM-OP3 was applied to attribute all secondary products to sources emitting $NO_x$ or reactive VOC species and radicals when present in the parent reactants, and otherwise assign them based on stoichiometric reaction rates (Shu et al., 2023).

**Comment 17**

Page 5, Line 12: Clarify whether the 2018 evaluation is conducted using 2018 SEIM and MEIC emissions. The referencing of years throughout the manuscript is inconsistent and may cause confusion.

**Response:**

Thanks for your comment. We confirm that the 2018 simulation was conducted using the 2018 versions of both the SEIM and MEIC emission inventories. Since the meteorological fields were fixed for the year 2018, the model evaluation was only performed for the 2018 simulation to ensure consistency across input datasets. We have revised the manuscript to clarify this point.

**Revisions in Main Text:**

**2.2 Air quality model**

We evaluated the simulated $O_3$ concentrations for the of 2018 against 1455 available ground-based observations (**Figure S3**) for model validation.

**Comment 18**

Table S3: As with Table S1, improve the figure caption by clarifying how the metrics are calculated. Consider computing statistics based on MDA8 values rather than hourly data.

**Response:**

Thanks for your comments. We have revised the caption by clarifying how the metrics are calculated. We agree that MDA8 O3 is a more relevant metric for evaluating ozone pollution and have accordingly recalculated the statistics using MDA8 values. The updated results have been incorporated into Table S3.

**Revisions in Main Text:**

Table S3. CMAQ performance (Number of stations:1455).

| Pollutants | Month | Mean OBS | Mean SIM | MB | NMB | NMGE | MFB | MFE | R |
|---|---|---|---|---|---|---|---|---|---|

| | | | | | | | | |
|---|---|---|---|---|---|---|---|---|
| | Winter | 40.53 | 37.57 | -2.69 | -0.05 | 0.60 | -0.39 | 0.82 | 0.58 |
| Maximum daily 8-hour average O$_3$ | Spring | 77.92 | 77.26 | 0.06 | 0.04 | 0.43 | -0.11 | 0.55 | 0.69 |
| | Summer | 69.24 | 79.15 | 9.67 | 0.19 | 0.51 | -0.01 | 0.57 | 0.69 |
| | Autumn | 37.60 | 43.08 | 5.50 | 0.19 | 0.74 | -0.18 | 0.83 | 0.58 |

* $MB = \frac{\sum_{i=1}^{N} S_i - O_i}{N}$, $NMB = \frac{\sum_{i=1}^{N} S_i - O_i}{\sum_{i=1}^{N} O_i}$, $NMGE = \frac{\sum_{i=1}^{N} |S_i - O_i|}{\sum_{i=1}^{N} O_i}$, $MFB = \frac{2 \times \sum_{i=1}^{N} \frac{S_i - O_i}{S_i + O_i}}{N}$, $MFE = \frac{2 \times \sum_{i=1}^{N} \frac{|S_i - O_i|}{S_i + O_i}}{N}$, $R = \frac{\sum_{i=1}^{N} [(S_i - \bar{S}) \times (O_i - \bar{O})]}{\sqrt{\sum_{i=1}^{N} (S_i - S)^2 \times \sum_{i=1}^{N} (O_i - \bar{O})^2}}$,

* $N$ is the total number of samples in the dataset, $S_i$ is the simulated value of the $i$-th sample, $O_i$ is the observed (measured) value of the $i$-th sample, $\bar{S}$ is the mean of the simulated values, $\bar{O}$ is the mean of the observed values.

**Comment 19**

Page 5, Lines 22–23: The Random Forest model is trained using CMAQ outputs, not ISAM. Including ISAM-derived information in the machine learning model could potentially enrich the analysis. Monthly averages may obscure key insights regarding the impact of shipping on peak ozone levels.

**Response:**

Thank you for the valuable comment. Our study focuses on exploring the relationship between emissions and O3 pollution from a broader, long-term perspective. The goal is to extract historical patterns that can inform future emission reduction strategies. While short-term O3 peaks are indeed important, our analysis is not aimed at capturing transient pollution episodes. In China, emission control policies are typically oriented toward long-term O3 mitigation. Short-term peak O3 control may not be effectively addressed through emission reductions from a single sector, and often requires coordinated meteorological and emergency response measures. Therefore, our findings are more aligned with supporting long-term planning rather than short-term event-specific interventions. We have clarified this research objective in the Introduction.

Moreover, the temporal resolution of the emission inventories used in this study is not sufficient to support hourly or daily analyses of ozone peaks. Specifically, the SEIM inventory provides annual totals for 2016–2019 and monthly totals for 2020, while MEIC offers monthly totals. Although hourly emissions were generated for CMAQ simulations through temporal allocation profiles, these are based on generalized assumptions and do not reflect real-time activity patterns. As such, any analysis of short-term O3 variability based on these inputs would be inherently uncertain and potentially misleading.

**Comment 20**

Page 5, Line 28: This sentence highlights a key concern. If the main purpose of the Random Forest model is as stated here, the ISAM module already provides more direct and robust information. Please revise the sentence and clarify the rationale for applying machine learning.

**Response:**

Thank you for your insightful comment. We agree that the ISAM module provides direct and valuable source apportionment results. However, as noted in the revised manuscript, ISAM-based assessments are limited by the finite number of predefined perturbation scenarios that can be feasibly simulated due to high computational costs. This constraint hampers the ability to explore nonlinear and complex source-receptor relationships across broader meteorological and emission variability. To address this limitation, we applied an explainable machine learning model trained on ISAM outputs to extend insights beyond the original scenarios. This approach enables the identification of key emission drivers and their nonlinear impacts on O3 formation, helping to uncover hidden patterns and mechanisms that are otherwise difficult to extract from a limited set of CTM simulations. We have revised the relevant sentence in the manuscript to better clarify this rationale.

**Revisions in Main Text:**

**2.3 Explainable machine learning model**

Although CMAQ-ISAM can generate large amounts of simulation data to investigate the impacts of shipping emissions, the number of scenarios remains limited due to computational constraints. As a result, current analyses struggle to elucidate the mechanisms by which

shipping emissions contribute to $O_3$ formation from these discrete scenarios. In particular, capturing nonlinear interactions between emission sources, meteorological conditions, and chemical processes is challenging when only a limited number of emission perturbations are available. Recently, the advancement of machine learning techniques, especially explainable models, has provided a promising complementary approach (Yao et al., 2024a; Yao et al., 2024b; Liu et al., 2025). These models can learn from existing models to approximate the source-receptor relationships embedded in the simulation results. By identifying key emission drivers, quantifying their nonlinear contributions to $O_3$, and revealing latent patterns across spatiotemporal scales.

**Comment 21**

Page 5, Line 30: If results from 2016–2019 are used for training, does this imply that model predictions discussed in the Results section refer exclusively to 2020 emissions? Please clarify.

**Response:**

Thank you for the question. As the Random Forest model was trained using data from 2016 to 2019, the learned mapping between input features and predicted O3 concentrations is based solely on these historical samples. Therefore, the interpretability analysis is also conducted based on the 2016–2019 simulation results. We have clarified it in the Main Text now.

**Revisions in Main Text:**

**2.3 Explainable machine learning model**

In order to identify the sensitivity and response relationship between prediction variables and results in the RF models, the SHapley Additive exPlanations (SHAP) technique, a game-theoretic framework introduced by Lundberg et al. (Lundberg et al., 2020; Lundberg and Lee, 2017), was employed to interpret the pattern learned from the 2016-2019 simulation data by the RF model using the Python scikit-learn library.

**Comment 22**

Page 6, Line 14: Indicate whether the averages are calculated from hourly ozone values or based on MDA8. Since ozone concentrations at night are often overestimated in models, using

hourly data may introduce bias. MDA8 is a more appropriate metric for evaluating daily peaks and understanding emission sensitivity. As noted earlier, presenting five-year averages under the current setup is potentially misleading.

**Response:**

We thank the reviewer for this valuable comment. As clarified, the averages in our study were calculated from hourly ozone values, not based on MDA8. We acknowledge that nighttime ozone may be overestimated in models and agree that MDA8 is more suitable for evaluating daily peaks and exposure-related assessments.

To assess the potential bias introduced by using hourly data, we conducted a comparison between shipping-related O3 contributions calculated using hourly means and MDA8 values across multiple months. Over oceanic regions, we found that the difference can reach 2−5 ppb, while over land, the bias is generally within 2 ppb. Given that our study focuses on multi-year and seasonal mean contributions, rather than short-term episodic events or exceedance-specific analysis, we retained the hourly-based averaging approach for consistency and computational feasibility.

Moreover, since the overall contribution of shipping emissions to ozone levels is relatively low, this difference is unlikely to significantly affect the conclusions. Some previous studies investigating shipping-related O3 have also adopted hourly or daily mean O3 without necessarily using MDA8 O3 (Fu et al., 2023; Wan et al., 2023).

Nonetheless, we fully recognize that this choice may introduce some uncertainty in quantifying peak-level responses, and we have explicitly acknowledged this limitation in the revised manuscript.

**Revisions in Main Text:**

**2.4 Limitations**

In this study, monthly and annual mean $O_3$ concentrations were derived from hourly model outputs, rather than the widely used MDA8 O3. While this approach is consistent with the

study's focus on long-term trends and average responses, it may introduce bias due to the well-known overestimation of nighttime ozone in chemical transport models. A sensitivity test comparing shipping-related $O_3$ contributions based on hourly averages and MDA8 revealed that over oceanic areas, the difference may reach 2-5 ppb, while over land, it remains within 2 ppb. Given that the relative contribution of shipping emissions to total $O_3$ is generally low, the impact of this bias is expected to be limited.

**Figure 3a** shows the five-year average of shipping-related $O_3$ calculated based on hourly values, which is defined as the sum of $O_3$ concentration caused by emissions of OGVs, CVs, and RVs traced by CMAQ-ISAM.

**Comment 23**

Page 6, Figure 3: Consider presenting total ozone concentrations and shipping contributions in separate panels using absolute values for clarity.

**Response:**

Thank you for your suggestion. We agree that presenting total O3 concentrations and shipping contributions in separate panels using absolute values may help isolate individual effects. However, our intention with Figure 3 was to specifically highlight the spatial distribution of shipping-related $O_3$ contributions, both in absolute and relative terms. Showing total O3concentrations alongside shipping-related contributions may reduce the clarity of the patterns attributed solely to shipping.

Moreover, total O3 concentrations are not the focus of our analysis and are not further interpreted in this study, as their characteristics and distributions have already been extensively addressed in many previous works. Our objective here is to isolate and interpret the influence of shipping emissions specifically, which we believe is better achieved by focusing the figure layout on shipping-related contributions alone.

We believe the current presentation more effectively serves the purpose of assessing the spatial heterogeneity and regional importance of shipping emissions on O3 pollution.

**Comment 24**

Page 7, Line 25: Please explain why the relative contribution of shipping emissions appears higher in 2017 than in subsequent years, despite steadily increasing emissions (as shown in Figure 1). This suggests regional sensitivity that warrants further discussion.

**Response:**

Thank you for your insightful comment. Upon reviewing the data across the full domain and key regions, we found that the highest relative contribution of shipping emissions to O3 pollution actually occurred in 201. Notably, the relative contributions in 2018 and 2020 were slightly lower than that in the previous year.

This discrepancy, where shipping NOx and VOC emissions increased steadily, but relative contributions to O3 did not follow a consistent upward trend, can be attributed to the complex and nonlinear relationship between emissions and O3 formation. As shown in Figure 9, an increase in shipping emissions does not necessarily result in a rise in O3 levels. Furthermore, declining land-based anthropogenic NOx emissions during this period may have promoted O3 formation, thereby amplifying the relative role of shipping emissions in certain years.

It is worth noting that while our subsequent explainable machine learning analysis (e.g., SHAP-based interpretation) can help explore how total O3 responds to different emission inputs. However, understanding changes in relative contributions would require predicting shipping-related O3 alone using RF. Given the relatively small magnitude of shipping-related O3, current RF models are limited in their ability to accurately reproduce such values. As a result, providing a precise explanation for interannual variability in relative contributions remains challenging and is an area for future methodological development.

We have also highlighted this interesting finding in the revised manuscript and proposed potential explanations to guide future investigations.

**Revisions in Main Text:**

**3.1 Annual O$_3$ impact from shipping emissions**

It is worth noting that, despite continuous increases in shipping $NO_x$ and VOC emissions, their relative contributions to $O_3$ decreased in 2018 and 2020. This pattern may result from simultaneous land-based emission reductions, which can affect atmospheric oxidizing capacity (Lv et al., 2020).

**Comment 25**

Page 8, Line 9: Include, in parentheses, the relative contribution of shipping to total ozone.

**Response:**

Thanks for your comment. We have added the Table S4 to provide this information.

**Revisions in Main Text:**

**3.2 Contribution of different types of vessels**

**Figure 5** and **Table S4** shows the five-year average contribution of emissions from different ship types to the shipping-related $O_3$ and the total $O_3$, respectively. Nationwide, OGVs, CVs, and RVs contributed 2.6%, 2.6%, and 3.3% to the total $O_3$, respectively.

**Revisions in Supplement:**

Table S4. Five-year average of total $O_3$ and shipping-related $O_3$ across China. (ppb)

|       | Total | All ships    | RVs         | CVs         | OGVs         |
|-------|-------|--------------|-------------|-------------|--------------|
| China | 40.77 | 3.49 (8.6%)  | 1.07 (2.6%) | 1.08 (2.6%) | 1.34 (3.3%)  |
| BRA   | 31.02 | 4.48 (14.4%) | 0.80 (2.6%) | 1.60 (5.2%) | 2.08 (6.7%)  |
| YRD   | 25.64 | 5.55 (21.6%) | 1.70 (6.6%) | 1.76 (6.9%) | 2.09 (8.2%)  |
| PRD   | 28.77 | 8.92 (31.0%) | 2.32 (8.1%) | 2.40 (8.3%) | 4.20 (14.6%) |
| IRA   | 41.23 | 3.95 (9.6%)  | 1.55 (9.6%) | 1.14 (9.6%) | 1.26 (9.6%)  |

\* The numbers in parentheses indicate the relative contribution.

**Comment 26**

Page 9, Line 17: To strengthen the analysis, consider presenting the full range (e.g., min, max, interquartile range) of shipping contributions, rather than just the mean. Contributions in

specific regions may be substantial.

**Response:**

Thanks for your comment. Thank you for your suggestion. We have added the full range of shipping-related O3, and substantially revised this section accordingly to highlight regional differences. Please see the updated content and Table 1 in the revised manuscript.

**Revisions in Main Text:**

**3.3 Seasonal $O_3$ impact from shipping emissions**

The five-years-average seasonal variations in the contribution of shipping emissions to $O_3$ concentrations across different regions are shown in **Figure 7** and **Tabel 1**, with January, April, July, and November representing winter, spring, summer, and fall, respectively. For cold seasons, including winter and fall, due to weaker solar radiation and lower temperatures that limit $O_3$ formation (**Figure S4**), the shipping-related $O_3$ remains relatively lower than warm seasons (spring and summer), with national average and relative contributionof 1.53 ppb (5.6%) and2.41 ppb (7.9%), respectively (**Figure S5**). However, in the south of PRD, especially Guangdong and Hainan Provinces (**Figure 7a**, **7d**, and **Table 1**), the average and maximum of seasonal shipping-related $O_3$ exceeds 5 ppb and 21 ppb, respectively, Notably, fall pollution even severer than that in summer. This is mainly because the PRD remains warm and humid in fall, and prevailing monsoon winds are more likely to transport ship-borne pollutants from the sea to inland areas (**Figure S4**, **S6**, and **S7**). Another distinct pattern is observed in BRA, where shipping-related $O_3$ formation tends to be more localized during the cold seasons, as indicated by a larger difference between the median and average values (**Table 1**). During this period, mainland China is under the influence of the Mongolian High Pressure System, and continental winds generally suppress the inland transport of ship-related $O_3$ (Cheng et al., 2023; Zhao et al., 2023). Therefore, significant shipping-related $O_3$ pollution only appears in major port cities with intensive maritime activity.

In spring, shipping-related $O_3$ reached its peak in YRD and PRD , with the maximum value exceeding 30 ppb (**Figure 7b** and **Table 1**), consistent with the results of previous studies (Cheng et al., 2023; Schwarzkopf et al., 2022). Although spring is generally less favorable for $O_3$ formation compared to summer in terms of temperature and humidity, strong onshore winds may play an important role in reduce the influence of shipping emissions (Cheng et al., 2023; Ma et al., 2022) (**Figure S4**, **S6**, and **S7**). In addition, more complex physicochemical interactions may drive springtime $O_3$ (Cao et al., 2024; Zhang et al., 2024), which needs further investigation. In summer, shipping emissions significantly increased $O_3$ concentrations nationwide by 4.77 ppb and responsible for 13.7% of national $O_3$ pollution (**Figure 7c** and **Figure S5**). Notably, even in IRA, where shipping emissions are much lower than in coastal regions, shipping-related $O_3$ were comparable to those along the coast. This is primarily because central China lies in a perennial monsoon region, where summer monsoons can carry shipping-related air pollutants inland from coastal cities (Zheng et al., 2024).

Table 1 Seasonal ranges of shipping-related $O_3$ across BRA, YRD, PRD, and IRA. (unit: ppb)

| Region | Metric | Winter | Spring | Summer | Fall |
|--------|--------|--------|--------|--------|------|
| BRA | Minimum | 0.01 | 0.83 | 1.25 | 0.06 |
| | 25% Quartile | 0.05 | 3.09 | 7.13 | 0.39 |
| | Median | 0.11 | 4.02 | 9.32 | 0.78 |
| | 75% Quartile | 0.91 | 6.55 | 11.90 | 1.63 |
| | Maximum | 6.59 | 23.22 | 32.40 | 14.39 |
| | Mean | 0.71 | 5.36 | 10.13 | 1.74 |
| YRD | Minimum | 0.20 | 1.92 | 1.45 | 0.55 |
| | 25% Quartile | 1.01 | 5.02 | 6.11 | 1.91 |
| | Median | 1.79 | 6.64 | 7.29 | 3.23 |
| | 75% Quartile | 2.93 | 8.35 | 9.03 | 5.14 |
| | Maximum | 16.32 | 31.47 | 25.86 | 24.59 |
| | Mean | 2.47 | 7.39 | 7.92 | 4.41 |
| PRD | Minimum | 1.11 | 3.91 | 0.11 | 1.91 |
| | 25% Quartile | 2.79 | 7.94 | 3.19 | 4.85 |
| | Median | 5.19 | 10.07 | 5.65 | 7.97 |
| | 75% Quartile | 7.68 | 15.17 | 7.77 | 11.25 |
| | Maximum | 21.98 | 33.46 | 26.19 | 28.78 |
| | Mean | 5.96 | 11.91 | 5.77 | 8.79 |
| IRA | Minimum | 0.17 | 0.91 | 2.32 | 0.10 |
| | 25% Quartile | 1.19 | 2.88 | 5.65 | 1.44 |
| | Median | 1.61 | 4.85 | 9.07 | 2.85 |
| | 75% Quartile | 2.29 | 5.73 | 11.31 | 4.16 |
| | Maximum | 5.53 | 11.91 | 26.19 | 7.52 |
| | Mean | 1.76 | 4.58 | 9.03 | 2.87 |

**Comment 27**

Page 11, Line 3: No comments are made on seasonal variations in emissions. Do RV or CV emissions exhibit any significant seasonal patterns?

**Response:**

Thank you for the comment. In our study, the annual shipping emissions from 2016 to 2019 were first estimated based on AIS data, and then temporally averaged to derive uniform monthly emissions for each year. In our previous work, where we found that the monthly variation in RV and CV activity levels was generally small, with only a slight decrease during the Spring Festival in winter. The summer fishing off-season mainly affects fishing vessels,

which contribute only marginally to total emissions (as shown in the Figure below).

For 2020, we used monthly emission inventories to reflect the impacts of COVID-19. However, since the interannual variation in emissions was limited, the seasonal analysis in this study was based on the 5-year average (2016–2020); therefore, the impact of monthly variations in emissions on seasonal patterns was not emphasized.

We have also revised Table S2 to clarify which inventories are monthly versus annual averages.

[Figure]

Figure Statistics of vessels' dynamic and static information for 2016–2019. (a) Daily average operating hours. (b) Vessel fleet compositions from different aspects. *Ref.* https://doi.org/10.5194/acp-21-13835-2021.

**Revisions in Supplement:**

**Table S2. Emissions used in the CMAQ model.**

| Emissions | Year | Reference |
|---|---|---|
| Shipping Emissions | 2016-2020

2016-2019: Monthly-averaged from annual totals

2020: Monthly | This study |
| Land-based anthropogenic emissions in China (mobiles, industry, power, domestic, | 2016-2020
Monthly | MEIC (http://www.meicmodel.o |

and agriculture)    rg/,

last access: November
2023)
* * *
**Comment 28**

Page 11, Line 5: The analysis presented may be incomplete due to the omission of hemispheric background ozone concentrations.

**Response:**

Thank you for the comment. We acknowledge that background $O_3$ concentrations are important for understanding surface ozone levels. However, they were not included in our model, and this omission does not affect the reliability of our results, for the following reasons:

SHAP is a relative explanatory framework rather than an absolute causal attribution tool. It evaluates the average marginal contribution of each input feature to the model prediction, based on all possible permutations within the existing feature space. Specifically, a SHAP value quantifies the change in model output before and after including a given feature, averaged across all feature combinations.

In our study, background $O_3$ concentrations were not included as model inputs, which may have some influence on overall model fitting performance. However, as shown in our model evaluation results, the prediction accuracy remains acceptable. More importantly, the omission of background $O_3$ does not distort the relative contribution rankings provided by SHAP, nor does it lead to incorrect attribution to the included features.

Therefore, the absence of background $O_3$ should not be interpreted as undermining the model's ability to explain the prediction outcome based on the existing input variables.

In addition, current studies employing machine learning explainability for $O_3$ pollution rarely include background concentrations, as summarized in Table.

| Input variables | Reference |
| --- | --- |

| | |
|---|---|
| Pollutant concentrations, meteorological data, and data related to regional transport | (Li et al., 2025) |
| Monthly trend of O3, wind direction, wind speed, radiation, temperature, evaporation, BLH, precipitation, total cloud cover, and pressure | (Yao et al., 2024b) |
| Atmospheric pollutants ($SO_2$, $NO_2$, CO, $PM_{2.5}$, $PM_{10}$, and $O_3$), meteorological parameters (T, RH, WS, SP, and WD), and temporal characteristics. | (Yao et al., 2024a) |
| Volatile organic compounds (VOCs, ppb), particulate matter ($PM_{10}$, $PM_{2.5}$, $PM_1$, TSP, $\mu g/m^3$), trace gases (NOx, NO, $NO_2$, $SO_2$, $O_3$, $\mu g/m^3$; CO, $mg/m^3$), particulate carbon (OC, EC, TC, $\mu g/m^3$), and meteorological variables, including air temperature (T, Cº), relative humidity (RH, %), wind speed and direction (WS and WD, m/s), solar radiation ($w/m^2$) and visibility (km) | (Wang et al., 2023) |

Finally, we also note that while our CMAQ-ISAM simulations account for the contributions from boundary conditions (BCON) and initial conditions (ICON), they do not provide explicit background O3 concentrations.

**Comment 29**

Page 13, Line 8: This paragraph appears to question the robustness of the machine learning approach for analyzing ozone formation. Consider clarifying its intended role and limitations in this context.

**Response:**

Thank you for the insightful comment. To improve clarity and ensure consistency, we have moved the original paragraph discussing the limitations of the machine learning approach to the Limitations section. In this revised context, we further clarified the intended role of SHAP-based interpretability, its dependence on input features, and its inability to reflect causal relationships. We also explicitly acknowledged the exclusion of background ozone

concentrations.

**Revisions in Main Text:**

**2.4 Limitations**

Explainable machine learning model relies on the structure and quality of the input dataset and cannot account for unmeasured or omitted variables, such as hemispheric background ozone concentrations. As a result, the derived feature importance reflects statistical associations rather than causal relationships. It should be noted that if one seeks to determine whether a given variable promotes or suppresses $O_3$ pollution using machine learning methods, additional field observations, experimental data, and corresponding simulation results may be required as supporting evidence. Considering the interactions among variables, even if individual contributions are small, the SHAP estimates for each explanatory variable are unlikely to perfectly reflect their actual contributions in the underlying physical processes. Furthermore, in the presence of strong collinearity or complex nonlinear interactions, SHAP values may not fully disentangle overlapping influences among features.

**Comment 30**

Page 13, Line 17: The conclusion section is currently too brief and does not convey the potential key findings of the study. Some conclusions (e.g., the role of temperature and solar radiation) are well known and may not constitute novel insights. The authors should more clearly explain the main findings and novelty of their work.

**Response:**

Thank you for the valuable suggestion. In response, we have thoroughly revised the Conclusion section to better highlight the potential key findings of our study. The updated conclusion now clearly summarizes the long-term trends, regional and seasonal characteristics of shipping-related O3 pollution, and the differentiated roles of various ship types. We also provide region-specific policy implications, including the importance of coordinated land-based and shipping emission controls, the need to address inland river vessel emissions, and the benefits of implementing seasonal and air quality-oriented management measures. These additions better

reflect the novelty and policy relevance of our work.

**Revisions in Main Text:**

**4 Conclusion**

In this study, we conducted multi-year CMAQ-ISAM simulations to investigate the how shipping emissions impacted $O_3$ across China, with a focus on three coastal regions and a inland region. From 2016 to 2020, shipping emissions increased national average $O_3$ concentrations by 3.5 ppb, accounting for 8.6% of total $O_3$, with a spatial gradient decreasing from coastal to inland regions. Despite the increasing intensity of shipping activity and the implementation of the global sulfur cap, shipping $NO_x$ and VOCs emissions rose significantly during this period. However, the national average shipping-related $O_3$ increased by only 0.23 ppb, while the relative contribution of shipping emissions to $O_3$ pollution rose by approximately 0.5%. Notably, this relative contribution did not increase continuously; instead, a decline was observed in 2018 and 2020. This non-linear response, under conditions of simultaneous changes in multiple pollutants from different sectors, highlights the complexity and need for further investigation of attribution of $O_3$ pollution. For the four focus regions, the contribution of shipping to $O_3$ levels exceeded the national average, with more pronounced interannual increases.

We further disaggregate ship types to OGVs, CVs, and RVs. The result revealed that OGVs were the dominant contributors to shipping-related $O_3$ in coastal areas, followed by CVs, whereas RVs were the main source in inland river areas. Although OGVs, CVs, and RVs differ significantly in their emission magnitudes, the difference in their contributions to $O_3$ pollution is gradually narrowing. This trend suggests that the influence of RVs on regional $O_3$ levels should no longer be overlooked and that emission control efforts for RVs deserve renewed attention. However, from the perspective of sulfur emission control, RVs in China had already reached the final stage of sulfur regulation by 2018 under the implementation of domestic emission control policies. In contrast, $NO_x$ control for inland vessels remains largely unaddressed. Globally, there is limited precedent or experience in regulating $NO_x$ emissions from inland waterways, leaving China without a clear reference framework for RVs $NO_x$

mitigation. Future control of shipping NO$_x$ emissions needs to take into account both inland waterways and coastal areas.

The impacts of shipping emissions on O$_3$ also exhibited significant seasonal and regional characteristics. While shipping-related O$_3$ levels were generally lower in colder seasons, fall pollution in southern coastal regions exceeded that of summer due to favorable land–sea monsoon transport. Peak shipping-related O$_3$ levels occurred in spring over YRD and PRD, and in summer over inland areas. These patterns highlight the importance of implementing seasonal and region-specific control strategies to mitigate shipping-related O$_3$ pollution effectively. In particular, quality-oriented management policies such as seasonal routing adjustments, port operation scheduling, or dynamic emission monitoring, may play a more immediate role than emission control policies, which are typically less adaptable to seasonal variability and require long-term infrastructure or regulatory changes. Therefore, combining flexible operational measures with long-term emission reduction plans could enhance the overall effectiveness of O$_3$ mitigation.

Interpretable machine learning analysis further revealed significant spatial differences in the contribution of shipping emissions to O$_3$. In BRA and IRA, O$_3$ formation was primarily driven by land-based NO$_x$ and VOC emissions, with shipping emissions playing a minor role and even showing a suppressive effect on O$_3$ formation. In contrast, in coastal regions such as YRD and PRD, the increasing share of shipping emissions in the total anthropogenic emissions enhanced their contribution to O$_3$, with shipping NO$_x$ emissions showing a slight promoting effect on O$_3$ formation. This regional difference suggests that solely controlling shipping emissions may lead to unexpected atmospheric chemical responses and, under certain conditions, could even cause an increase in O$_3$ concentrations. Therefore, effective O$_3$ pollution control requires a coordinated reduction of both land-based and shipping emissions, based on regional emission structures and atmospheric oxidation characteristics.

**Technical Comments**

**Comment 1**

The quality of several figures should be improved for readability and clarity.

**Response:**

Thank you for your comment. We will upload high-resolution vector versions of the figures separately to ensure readability and clarity.

**Comment 2**

All figure and table captions should be self-contained and descriptive, clearly explaining the data presented.

**Response:**

Done.

**Comment 3**

Page 1, Line 17: Replace "...mechanisms of shipping emissions..." with "...mechanisms by which shipping emissions...".

**Response:**

Done.

**Revisions in Main Text:**

…and explore mechanisms by which shipping emissions influence $O_3$ formation.

**Comment 4**

Page 2, Line 16: "volatile organic compounds"

**Response:**

Done.

**Revisions in Main Text:**

Ships emit both gaseous and particulate pollutants, including sulfur dioxide ($SO_2$), nitrogen oxides ($NO_x$), particulate matter, and volatile organic compounds (VOC).

**Comment 5**

Page 2, Lines 22–23: "critically important"

**Response:**

Done.

**Revisions in Main Text:**

Therefore, clarifying the historical and current contribution of shipping emissions to the formation of $O_3$ is critically important for further pollution control in China.

**Comment 6**

Page 2, Line 34: Ensure consistent terminology throughout the manuscript. Use either "ozone" or "$O_3$," not both interchangeably.

**Response:**

We have reviewed the entire manuscript and ensured consistent terminology by using "$O_3$" throughout the text. The specific changes are not listed here for brevity.

**Comment 7**

Page 2, Line 35: Replace "timeframes" with "periods."

**Response:**

Done.

**Revisions in Main Text:**

Furthermore, previous studies were limited to restricted periods.

**Comment 8**

Page 3, Line 3: The sentence is unclear; please revise for clarity and correct any typographical errors.

**Response:**

We have re-written this sentence.

**Revisions in Main Text:**

Furthermore, although model-based assessments can generate large amounts of simulation data to investigate the impacts of shipping emissions, the number of scenarios that can be simulated by chemical transport models remains limited due to computational constraints. As a result, current analyses struggle to struggles the mechanism of how shipping emissions contribute to $O_3$ formation from these discrete scenarios. Recently, the advancement of machine learning techniques, with strong capabilities in capturing nonlinear relationships, provides a valuable approach for uncovering underlying patterns in such datasets (Luo et al., 2025).

**Comment 9**

Page 3, Line 8: Replace "allocate culpabilities of" with "apportion"

**Response:**

Done.

**Revisions in Main Text:**

We also apportion the contribution of shipping emissions from ocean-going vessels (OGVs), coastal vessels (CVs), and river vessels (RVs) to $O_3$ pollution to identify the influences of regionally differentiated shipping emission control policies.

**Comment 10**

Page 3, Line 18: Would not Wang et al. (2021) be the appropriate reference for SEIMv2.0?

**Response:**

We have corrected the reference.

**Revisions in Main Text:**

The Shipping Emission Inventory Model (SEIM v2.0) is a disaggregate dynamic method (Wang et al., 2021) driven by…

**Comment 11**

Page 3, Line 18: Remove the word "driven" after "by."

**Response:**

Done

**Comment 12**

Page 3, Line 30: Use "VOC" instead of "HC."

**Response:**

Done

**Revisions in Main Text:**

In the SEIM, shipping emissions for both air pollutants (e.g., $SO_2$, PM, $NO_x$, CO and VOC) and greenhouse gases (e.g., $CO_2$, $CH_4$ and $N_2O$) from the main engines, auxiliary engines and boilers were calculated, detailed information of SEIM is described in our previous study (Wang et al., 2021).

**Comment 13**

Page 3, Line 38: Replace "IMO." with "IMO;"

**Response:**

Done

**Revisions in Main Text:**

OGVs were identified by both valid International Maritime Organization (IMO) numbers and the Maritime Mobile Service Identity (MMSI) numbers, since they are mostly engaged in international trade following the management of the IMO;

**Comment 14**

Page 3, Line 40: Replace "RVs. (c) Finally, vessels" with "RVs; and (c) vessels."

**Response:**

Done

**Revisions in Main Text:**

Vessels with more than 50 % of the AIS signals throughout the entire year occurring on inland rivers (14–43° N, 104–130° E) were considered as RVs; and (c) vessels that are not identified as OGVs or RVs are regarded as CVs.

**Comment 15**

Page 4, Line 15: Define the acronyms BRA, YRD, and PRD at first mention.

**Response:**

Thanks for your comments. BRA, YRD, and PRD are defined in the Introduction.

**Comment 16**

Page 5, Line 8: Correct the citation typo.

**Response:**

Done

**Revisions in Main Text:**

the open burning emissions from Cai's study (Cai et al., 2017).

**Comment 17**

Page 6, Figure 2: Clarify what is plotted. Does each point represent the monthly average per grid cell?

**Response:**

Done

**Revisions in Main Text:**

**Figure 2.** Performances of RF models for (a) BRA, (b) YRD, (c) PRD and (d) IRA. Each point represents the monthly average $O_3$ concentration at each CMAQ grid cell.

**Comment 18**

**Response:**

Done

**Revisions in Main Text:**

In contrast, although the BRA is also a coastal region, , it experiences lower temperatures and weaker solar radiation (Figure S4).

**Comment 19**

Page 12, Line 11: The quality of the circular plot is too low to read the percentage values. The figure caption should explain the plot clearly, including the meaning of horizontal and vertical displacement in the cloud of points for each feature.

**Response:**

Thank you for the suggestion. We have now added a detailed explanation of the circular plot in the revised figure caption, including the interpretation of both the horizontal and vertical displacement of points for each feature. We will upload a separate vector version of the figure to ensure clarity of the percentage values.

**Revisions in Main Text:**

**Figure 8.** Feature importance results of the random forest regression model for (a) BRA, (b) YRD, (c) PRD, and (d) IRA. The x-axis shows SHAP values representing the impact of each feature on $O_3$ predictions (positive: increasing $O_3$; negative: decreasing $O_3$). Each dot is a grid-month sample, with color indicating the feature value. Instances with identical x-values are stacked, and the stack height signifies the density.

**Comment 20**

**Response:**

Done

**Revisions in Main Text:**

Although OGVs, CVs, and RVs exhibit significant differences in their emissions.

**Comment 21**

Page 14, Line 4: Correct the typographical error.

**Response:**

We have carefully checked the sentence on Page 14, Line 4, but did not identify any typographical error. If possible, we would appreciate further clarification to ensure we address your concern accurately.

**References**

Cao, T., Wang, H., Chen, X., Li, L., Lu, X., Lu, K., Fan, S., 2024. Rapid increase in spring ozone in the Pearl River Delta, China during 2013-2022. npj Climate and Atmospheric Science 7, 309.

Cheng, Q., Wang, X., Chen, D., Ma, Y., Zhao, Y., Hao, J., Guo, X., Lang, J., Zhou, Y., 2023. Impact of Ship Emissions on Air Quality in the Guangdong-Hong Kong-Macao Greater Bay Area (GBA): With a Particular Focus on the Role of Onshore Wind. Sustainability 15, 8820.

Feng, X., Ma, Y., Lin, H., Fu, T.-M., Zhang, Y., Wang, X., Zhang, A., Yuan, Y., Han, Z., Mao, J., 2023. Impacts of ship emissions on air quality in Southern China: opportunistic insights from the abrupt emission changes in early 2020. Environmental Science & Technology 57, 16999-17010.

Fu, X., Chen, D., Wang, X., Li, Y., Lang, J., Zhou, Y., Guo, X., 2023. The impacts of ship emissions on ozone in eastern China. Science of the Total Environment 903, 166252.

IMO, 2018. Resolution MEPC.305(73), Amendments to the Annex of the Protocol of 1997 to Amend the International Convention for the Prevention of Pollution from Ships.

Li, Z., Bi, J., Liu, Y., Hu, X., 2025. Forecasting O3 and NO2 concentrations with spatiotemporally continuous coverage in southeastern China using a Machine learning approach. Environment International 195, 109249.

Liu, J., Chen, M., Chu, B., Chen, T., Ma, Q., Wang, Y., Zhang, P., Li, H., Zhao, B., Xie, R., 2025. Assessing the Significance of Regional Transport in Ozone Pollution through Machine Learning: A Case Study of Hainan Island. ACS ES&T Air 2, 416-425.

Liu, Y., Geng, G., Cheng, J., Liu, Y., Xiao, Q., Liu, L., Shi, Q., Tong, D., He, K., Zhang, Q., 2023. Drivers of increasing ozone during the two phases of clean air actions in China 2013–2020. Environmental Science & Technology 57, 8954-8964.

Lv, Z., Wang, X., Deng, F., Ying, Q., Archibald, A.T., Jones, R.L., Ding, Y., Cheng, Y., Fu, M., Liu, Y., 2020. Source–receptor relationship revealed by the halted traffic and aggravated haze in Beijing during the COVID-19 lockdown. Environmental science & technology 54, 15660-

15670.

Shu, Q., Napelenok, S.L., Hutzell, W.T., Baker, K.R., Henderson, B.H., Murphy, B.N., Hogrefe, C., 2023. Comparison of ozone formation attribution techniques in the northeastern United States. Geoscientific model development 16, 2303-2322.

Wan, Z., Cai, Z., Zhao, R., Zhang, Q., Chen, J., Wang, Z., 2023. Quantifying the air quality impact of ship emissions in China's Bohai Bay. Marine Pollution Bulletin 193, 115169.

Wang, H., Gao, Y., Sheng, L., Wang, Y., Zeng, X., Kou, W., Ma, M., Cheng, W., 2022. The impact of meteorology and emissions on surface ozone in Shandong Province, China, during summer 2014–2019. International Journal of Environmental Research and Public Health 19, 6758.

Wang, L., Zhao, Y., Shi, J., Ma, J., Liu, X., Han, D., Gao, H., Huang, T., 2023. Predicting ozone formation in petrochemical industrialized Lanzhou city by interpretable ensemble machine learning. Environmental Pollution 318, 120798.

Wang, R., Tie, X., Li, G., Zhao, S., Long, X., Johansson, L., An, Z., 2019. Effect of ship emissions on O3 in the Yangtze River Delta region of China: Analysis of WRF-Chem modeling. Science of The Total Environment 683, 360-370.

Wang, X., Yi, W., Lv, Z., Deng, F., Zheng, S., Xu, H., Zhao, J., Liu, H., He, K., 2021. Ship emissions around China under gradually promoted control policies from 2016 to 2019. Atmos. Chem. Phys. 21, 13835-13853.

Yao, L., Han, Y., Qi, X., Huang, D., Che, H., Long, X., Du, Y., Meng, L., Yao, X., Zhang, L., 2024a. Determination of major drive of ozone formation and improvement of O3 prediction in typical North China Plain based on interpretable random forest model. Science of The Total Environment 934, 173193.

Yao, T., Lu, S., Wang, Y., Li, X., Ye, H., Duan, Y., Fu, Q., Li, J., 2024b. Revealing the drivers of surface ozone pollution by explainable machine learning and satellite observations in Hangzhou Bay, China. Journal of Cleaner Production 440, 140938.

Yi, W., He, T., Wang, X., Soo, Y.H., Luo, Z., Xie, Y., Peng, X., Zhang, W., Wang, Y., Lv, Z., He, K., Liu, H., 2024. Ship emission variations during the COVID-19 from global and continental perspectives. Science of The Total Environment 954, 176633.

Zhang, X., Lu, X., Wang, F., Zhou, W., Wang, P., Gao, M., 2024. Enhanced Late Spring Ozone in Southern China by Early Onset of the South China Sea Summer Monsoon. Journal of Geophysical Research: Atmospheres 129, e2023JD039029.

---

## Author Comment (AC2)

**Response to Reviewer #1's Comments**

**General comment**

The paper treats of ozone formation trend (2016-2020) due to shipping emission in China by using modelling simulations suggesting the relevance of this source on this pollutant. The topic is interesting and suitable for the Journal. However, some aspects related to the choice done in modelling and to the interpretation of results are not completely clear or well described, see my specific comments. For this reason, I suggest considering the paper for publication after a revision step.

**Response:**

Thank you for your overall assessment and constructive suggestions. We appreciate your recognition of the relevance and timeliness of our study. In response to your comments, we have carefully revised the manuscript to clarify the modeling choices, refine the interpretation of the results, and address the specific concerns you raised. We hope the updated version more clearly conveys the scientific rationale, methodological robustness, and policy relevance of our work.

**Specific comments**

**Comment 1**

Anthropogenic emissions from other countries within the modeling domain (Table S2) was taken at 2010. It is possible to have a relevant uncertainty from this considering the period span of the study (2016-2020)?

**Response:**

Thank you for pointing out this important issue. We acknowledge that the use of anthropogenic emissions from other countries for the year 2010 could indeed introduce some uncertainty, particularly in boundary areas or regions with strong cross-border transport.

However, our primary focus is on the impacts of domestic shipping emissions within China, and most of the key regions of interest, such as the Yangtze River Delta, Pearl River Delta, Bohai Rim Area, and inland river areas, are less affected by boundary inflows from other

countries. In addition, our previous studies have demonstrated that this approach remains acceptable for regional simulations in China (Lv et al., 2020; Luo et al., 2024).

Furthermore, as shown in Table S3, the simulated O3 concentrations agree well with ground-based observations, which supports the reliability and acceptability of our model results despite this potential limitation.

**Revisions in Main Text:**

**2.4 Limitations**

Anthropogenic emissions from other countries within the modeling domain were held fixed at 2010 levels, and open burning emissions were fixed at 2015 levels throughout the simulation period (2016–2020). Although this assumption simplifies the modeling framework and is unlikely to significantly alter the relative changes in shipping-related $O_3$ assessed in this work, it may still introduce some degree of uncertainty, particularly in regions where long-range transport or fire-related emissions could have contributed more dynamically during specific years. Future studies could benefit from incorporating temporally varying background emissions to further reduce potential uncertainties and improve the representation of external influences.

**Comment 2**

Page 3, lines 1-4. It should be mentioned that there are also effects of titration of ozone due to ship emissions especially at local scale, a few kilometres, that could complicate both simulation and data interpretation see Merico et al (Atmospheric Environment 139, 2016, 1-10).

**Response:**

Thanks for your comment. we have now added a discussion of this effect in the Introduction.

**Revisions in Main Text:**

**1 Introduction**

Additionally, the titration of $O_3$ by NO from shipping emissions, particularly within a few kilometers of ship tracks, can further complicate the simulation and interpretation of $O_3$

concentrations at the local scale (Merico et al., 2016).

**Comment 3**

Page 3, line 6. Is this a sufficient resolution to investigate local processes leading to ozone formation? Generally, modelling of these processes is done using a much more refined scale.

**Response:**

Thanks for your question.

We agree that a finer spatial resolution is generally more appropriate for capturing local-scale ozone formation processes. However, our objective in this study is to assess the regional and interannual impacts of shipping emissions on ozone pollution at the national scale, rather than focusing on local photochemical processes at the urban or neighborhood level.

Therefore, the selected resolution of 36 km × 36 km represents a practical compromise between spatial detail and computational feasibility, especially considering the need to simulate multi-year scenarios (2016–2020) across the entire Chinese domain. This spatial resolution is also consistent with a series of studies by Geng et al. (as shown in the table below), who have extensively investigated ozone pollution and its driving mechanisms in China using similar model setups. We have added a statement in the Methods section.

| Reference | Model/Spatial resolution |
|---|---|
| Drivers of Increasing Ozone during the Two Phases of Clean Air Actions in China 2013–2020 | WRF-CMAQ/36 km |
| Evaluating the spatiotemporal ozone characteristics with high-resolution predictions in mainland China, 2013–2019 | WRF-CMAQ/36 km |
| Estimating Spatiotemporal Variation in Ambient Ozone Exposure during 2013–2017 Using a Data-Fusion Mode | WRF-CMAQ/36 km |

Additionally, the spatial resolution of the ship emission inventory we constructed is 0.05°, the land-based anthropogenic emission inventory from MEIC has a spatial resolution of 0.25°. Allocating land-based anthropogenic emissions to a much finer grid could significantly

increase the uncertainty of the simulation.

**Revisions in Main Text:**

**2.4 Limitations**

In this study, the spatial resolution of 36 km × 36 km may not fully capture the fine-scale spatial heterogeneity of $O_3$ concentrations, particularly in coastal urban areas where emissions and photochemical reactions exhibit strong spatial variability. This resolution is relatively coarse for accurately representing $O_3$ exceedances and local photochemical processes, which often occur at much finer spatial scales. Consequently, localized $O_3$peaks and gradients may be underestimated or smoothed in the model outputs. Despite this limitation, the selected resolution represents a practical compromise that enables multi-year simulations across the national domain.

**Comment 4**

Page 3, lines 31-32. What is Nm, nautical miles? Better to write it explicitly being not a SI unit.

**Response:**

Done.

**Revisions in Main Text:**

**2.1 Shipping emissions**

Here, emissions beyond 200 nautical miles from the Chinese mainland's territorial sea baseline were excluded from the domain by applying GIS-based spatial processing to the global shipping emission inventory, and only the annual shipping emissions from 2016 to 2020 within 200 nautical miles were used in the CMAQ-ISAM simulation.

**Comment 5**

The emissions used here, include the changes due to the implementation of IMO2020? It should be mentioned if it is expected an impact of this regulation on ozone formation due to shipping.

**Response:**

Thanks for your questions. The shipping emissions used in this study do account for the implementation of the IMO 2020 regulation.

Regarding the potential impact of IMO 2020 on ozone formation, although the regulation directly targets SO2 and PM emissions, its indirect effects on O3 may arise from increased VOC emissions. This is because low-sulfur fuels are typically richer in short-chain hydrocarbon (Wu et al., 2020). We have added a clarification in the manuscript to acknowledge this potential effect, although a detailed quantification of IMO 2020 impacts on $O_3$ formation is beyond the scope of this study and would require dedicated scenario analysis.

**Revisions in Main Text:**

**2.1 Shipping emissions**

Additionally, following the implementation of the global sulfur cap (IMO, 2018), the shift to low-sulfur fuels, which are typically richer in short-chain hydrocarbons (Wu et al., 2020), has contributed to a rise in shipping VOC emissions.

**3.1 Annual $O_3$ impact from shipping emissions**

**Figure 4** illustrates the interannual trend in shipping-related $O_3$ in key regions from 2016 to 2020. Nationwide, the shipping-related $O_3$ shows a slight upward trend, with an average annual growth rate of 1.7%, primarily observed in coastal regions. This trend aligns with the changes in shipping $NO_x$ and VOC emissions, especially in 2020 when a 0.2-0.3 ppb rise in shipping-related $O_3$ was observed, partly attributable to the notable increase in VOC emissions following the implementation of the global sulfur cap.

**Comment 6**

Page 4, line 18. Field rather than filed. In addition, why to use a one-year meteorology instead of the specific meteorology of each year? I believe that meteorological parameters have a strong influence on ozone formation and this is also what is mentioned in the conclusions..

**Response:**

Thanks for your comments. We have revised the typo error.

In this study, we primarily delve into the historical perspective of how anthropogenic emission changes impact shipping-related $O_3$. Consequently, we fixed the meteorological conditions to exclude their effects. We have now explained the reason for "fixing meteorological conditions" in the **2.2 Air quality model**.

Moreover, the impact of meteorological conditions should be insignificant. According to the National Climate Data Center (NCDC, ftp://ftp.ncdc.noaa.gov/pub/data/noaa/), the meteorological data (e.g., air temperature, relative humidity, monsoon) for the study area from 2016 to 2020 remained relatively stable (as shown in the Figure below). Additionally, based on the "China Climate Bulletin for the Year 2018", the climate conditions in China for the year 2018 were overall normal, with few extreme weather events, making it a representative meteorological year. Therefore, we fixed the annual meteorological conditions in the year 2018. Furthermore, although there may have been some extreme weather events during that year, our focus on interannual $PM_{2.5}$ variation minimizes the impact of these events.

[Figure]

Figure The meteorological conditions for CBS, SEC, SC and IRD for 2016 to 2020

**Revisions in Main Text:**

**2.2 Air quality model**

Here, we primarily focused on examining the impact of anthropogenic emission changes on

shipping-related $O_3$ from a historical perspective. To eliminate the impact of interannual meteorological variability, we used meteorological field of 2018 (Zhao et al., 2022), which simulated by WRF and identified as a typical meteorological year due to its relatively stable climate conditions, to drive the CMAQ simulations for the period 2016-2020.

**Comment 7**

Page 7, lines 25-26. This sentence seems to say that shipping is not relevant for ozone formation and it is opposite to what is said in conclusions.

**Response:**

Thank you for pointing this out. We agree that the original sentence could be misinterpreted as suggesting that shipping emissions are not relevant to O3 formation, which is not our intended meaning. Our point was that O3 responds to precursor changes in a nonlinear variable manner, and the shipping-related O3 increases are not directly proportional to the rise in shipping NOx and VOC emissions. We have revised the sentence to clarify this and avoid confusion with the conclusion section.

**Revisions in Main Text:**

**3.1 Annual $O_3$ impact from shipping emissions**

This is because the formation of $O_3$ depends on photochemical reactions involving $NO_x$ and VOC under solar radiation, and is influenced not only by the level of shipping emissions but also by land-based anthropogenic emissions, meteorological conditions, and long-range transport (Ye et al., 2023). Therefore, changes in shipping-related $O_3$ do not scale linearly with the changes in shipping $NO_x$ and VOC emissions.

**Comment 8**

Figure 1. What is the cause of the increment of emission in 2020? Fig. S2 does not show a significant increase of cargo throughput. Could it be simply related to the use of a different emission database?

**Response:**

We appreciate the reviewer's comment. The emissions in 2020 were estimated using a consistent emission database and methodology across all years, ensuring comparability. While Figure S2 shows that cargo throughput did not increase substantially in 2020, the emission increment is likely driven by a combination of factors beyond throughput alone. These include changes in vessel operating conditions (e.g., increased idling time), variations in ship traffic patterns, and potentially longer operating durations of high-emitting vessels. We presented cargo throughput as a straightforward proxy, but acknowledge that it may not fully capture the complex dynamics influencing emissions. A more in-depth investigation would be needed to disentangle the contributing factors, which is beyond the scope of this study. Nonetheless, we have added a brief explanation of this complexity in the revised manuscript to provide additional context.

**Revisions in Main Text:**

**2.1 Shipping emissions**

It is worth noting that changes in vessel operating conditions, such as idling time and engine load, also influenced emissions.

**Comment 9**

Page 14, line 4 there is an "s" that should be eliminated..

**Response:**

Thank you for your careful reading. We carefully checked the sentence on Page 14, Line 4, but we were unable to identify an extra or incorrect use of "s" in that line.

**References**

IMO, 2018. Resolution MEPC.305(73), Amendments to the Annex of the Protocol of 1997 to Amend the International Convention for the Prevention of Pollution from Ships.

Luo, Z., Lv, Z., Zhao, J., Sun, H., He, T., Yi, W., Zhang, Z., He, K., Liu, H., 2024. Shipping-related pollution decreased but mortality increased in Chinese port cities. Nature Cities 1, 295-304.

Lv, Z., Wang, X., Deng, F., Ying, Q., Archibald, A.T., Jones, R.L., Ding, Y., Cheng, Y., Fu, M., Liu, Y., 2020. Source–receptor relationship revealed by the halted traffic and aggravated haze in Beijing during the COVID-19 lockdown. Environmental science & technology 54, 15660-15670.

Merico, E., Donateo, A., Gambaro, A., Cesari, D., Gregoris, E., Barbaro, E., Dinoi, A., Giovanelli, G., Masieri, S., Contini, D., 2016. Influence of in-port ships emissions to gaseous atmospheric pollutants and to particulate matter of different sizes in a Mediterranean harbour in Italy. Atmospheric Environment 139, 1-10.